# Clinical study outcomes in IgA nephropathy: A systematic literature review and narrative synthesis

Anushya Jeyabalan[1]*, Kenar D. Jhaveri[2], Martin Bunke[3], Jonathon A. Briggs[4], David M.W. Cork[4], Mark E. Bensink[5]

1 Vasculitis & Glomerulonephritis Center, Massachusetts General Hospital, Boston, Massachusetts, United States of America, 2 Glomerular Center at Northwell Health, Division of Kidney Diseases and Hypertension and Donald and Barbara Zucker School of Medicine at Hofstra/Northwell, Great Neck, New York, United States of America, 3 C M Bunke Consulting, Mt Pleasant, South Carolina, United States of America, 4 Genesis Research, Newcastle upon Tyne, United Kingdom, 5 Travere Therapeutics, Inc., San Diego, California, United States of America

* ajeyabalan@mgh.harvard.edu

## Abstract

### Introduction

IgA nephropathy (IgAN) is an inflammatory kidney disease which, if left untreated, often progresses to kidney failure (KF). This systematic literature review identifies, collates, summarizes, and assesses the quality of clinical trial data describing the efficacy of therapies used for IgAN.

### Methods

Ovid Embase, PubMed, CENTRAL, and the Cochrane database of systematic reviews were searched on October 18th, 2021, and updated on December 12th, 2023. Electronic searches were supplemented with manual searches of key conferences, clinical trial registries, and bibliography screening. PRISMA and Cochrane guidelines were followed.

### Results

A total of 6710 references were identified (electronic and manual searches), of which 6483 were excluded. This resulted in 254 references reporting 183 studies which met our inclusion criteria. The majority of these IgAN studies (98/183 studies [60%]) had a non-randomized or single-arm design and/or a small population size or focused on dietary and traditional medicine, resulting in a high risk of bias and necessitated additional filtering to prioritize larger (n>30) randomized assessment of pharmacological interventions reporting key clinical outcomes. This additional filtering resulted in 76 randomized controlled trials (100 references) selected for narrative synthesis;

**Data availability statement:** All relevant data are within the paper and its Supporting Information files.

**Funding:** This work was funded by Travere Therapeutics.

**Competing interests:** AJ has served on a scientific advisory board for Callidftas Therapeutics. KDJ is a founder and co-president of the American Society of Onco-Nephrology; reports consultancy agreements with Secretome, George Clinicals, PMV pharmaceuticals and Callidtas. KDJ reports honoraria from the American Society of Nephrology, the ISN, and UpToDate.com; reports serving on the editorial boards of American Journal of Kidney Diseases, CJASN, Clinical Kidney Journal, Journal of Onconephrology, Kidney International, and Nephrology Dialysis Transplantation; reports serving as Editor-in-Chief of ASN Kidney News and section editor for onconephrology for Nephrology Dialysis Transplantation. MB is a consultant for Travere Therapeutics, Inc. JAB is an employee, and DMWC was an employee, of Genesis Research Group which received compensation from Travere Therapeutics, Inc. for conducting this study. MEB is a consultant for Travere Therapeutics, Inc. and reports an additional consultancy agreement with Amgen, Inc. This does not alter our adherence to PLOS ONE policies on sharing data and materials.

60 reported proteinuria outcomes and 18 reported estimated glomerular filtration rate (eGFR) outcomes.

## Conclusions

Until recently, the evidence has been mixed or inconsistent across studies for the efficacy of IgAN treatments in reducing proteinuria or slowing eGFR decline due to a high risk of bias in many included studies. The latest large, phase 3 NeflgArd (NCT03643965) and PROTECT (NCT03762850) clinical trials have demonstrated a meaningful reduction in proteinuria and eGFR decline for patients with IgAN receiving targeted-release formulation budesonide (TRF-B) or sparsentan. Results from other high-quality randomized controlled trials with a follow-up period of at least 2 years are still required to better support advancements in the management of IgAN.

## Introduction

IgA nephropathy (IgAN) is a progressive disease [1] and, if untreated, is a major cause of kidney failure (KF) (previously termed end-stage kidney disease) with considerable impacts on patients due to physical symptoms, chronic pain, and fatigue [2-4]. Indeed, in a cohort of IgAN patients KF or death occurred in 50% of patients during a median follow-up of 9.5 years [5].

IgAN therapy currently aims to preserve kidney function through management of blood pressure and proteinuria, which is pivotal in slowing progression to KF [6]. Initial therapy with either an angiotensin-converting enzyme inhibitor (ACEi) or angiotensin receptor blocker (ARB) is recommended in current KDIGO 2021 guidelines [6], and corticosteroid therapy is recommended for selected patients who remain at high risk of progressive kidney disease despite maximal supportive care[6]. These currently recommended treatments are non-targeted and used off-label with the aim of controlling symptoms and slowing progression to KF [7] resulting in a high unmet clinical need due to limited long-term impact on proteinuria [8,9] or long-term stabilization of eGFR [10]. ACEi/ARB therapies are less likely to be tolerated in older patients and patients with more severe disease, with higher baseline proteinuria and lower baseline eGFR [11]. Corticosteroid therapy for IgAN is also associated with a significant risk of severe adverse effects [6,12], including increased risk of diabetes mellitus, severe or fatal infection and osteonecrosis of the femoral head or bone fracture, particularly in older patients or patients with hypertension [13].

More recently, two new treatments have been approved for patients with IgAN in the US and Europe. Targeted-release formulation budesonide (TRF-B) (TARPEYO® [US] Kinpeygo® [Europe]), [14-17] and Sparsentan (FILSPARI®) [18-22].

This systematic literature review (SLR) identified clinical trials assessing treatments for IgAN and summarized key data in the form of a narrative synthesis to provide an overview of the current clinical trial evidence base. The narrative synthesis focused on RCTs for pharmacological therapies (supportive, immunosuppression/

immunomodulatory, combination, and non-immunosuppressive therapies) which report proteinuria and/or eGFR outcomes.

## Methods

### Data review methods and data sources

This SLR identified literature reporting clinical trials for IgAN treatments and was conducted in accordance with guidance from PRISMA [23], the Cochrane Handbook for Systematic Review of Interventions [24], and the Centre for Reviews and Dissemination [25]. The review protocol was not registered, and no amendments were made once the review was initiated.

Key literature databases (Ovid Embase, PubMed, CENTRAL, and the Cochrane database of systematic reviews) were searched on October 18th, 2021, and repeated on December 12th, 2023, to capture recent references. Additional searches were used to supplement the electronic database searches including screening the bibliographies from SLRs and meta-analyses which met the population inclusion criteria, as well as from included studies. Searches were also conducted of ClinicalTrials.gov, the International Clinical Trials Registry Platform (ICTRP) and conference abstracts from 2019-2023 (American Society of Nephrology [ASN], European Renal Association-European Dialysis and Transplant Association [ERA-EDTA], International Society of Nephrology [ISN], UK Kidney Week, and National Kidney Foundation meetings). All reference screening and data extraction were undertaken by 2 independent reviewers, with final decisions on study inclusion being confirmed by a third reviewer if required.

### Search strategy, study selection and data extraction

The scope of the review was defined using the patient, intervention, comparator, outcome, and study design (PICOS) framework (Table 1). Briefly, studies were required to include a population of patients of any age with IgAN or to report outcomes specifically for patients with IgAN within a mixed population (P); to assess any treatment (I) with any or no comparator (C); and studies assessing any efficacy or safety outcomes (O) were eligible for inclusion. Prospective clinical trials (Phase 1-4) (S) published from 1980 to 2023 were included. Peer-reviewed articles, conference presentations, and conference abstracts published in English were included. Where multiple publications reporting a single study are identified, the publications were grouped by study and data was extracted first from the full journal article. Where data were missing or incomplete, the other publications were screened for that information using the most recent publications and working backwards in time. Retrospective and observational studies were excluded.

Studies reporting outcomes for non-pharmacological interventions (e.g., dietary and lifestyle changes, traditional medicine, surgery) were excluded and hence data was not extracted nor assessed for risk of bias. Additional filtering was performed to select studies with the highest methodological quality for narrative synthesis (Table 1). This included selection of RCTs with more than 30 patients in the overall population. Additional filtering also focused on selecting studies which had assessed proteinuria (urinary protein creatinine ratio [PCR] or 24h urinary protein excretion rate [24h-PER]) or eGFR outcomes.

The search strings for Ovid Embase, PubMed and Cochrane databases are presented in S1–S3 Tables. The screening results of all studies identified in the literature search are available in S1 Data.

### Assessment of quality and risk of bias

Risk of bias was assessed at the study level according to the risk of bias assessments recommended by the NICE Single technology appraisal: User guide for company evidence submission template [26] for RCTs and the Newcastle-Ottawa scale for non-randomized studies for studies with a non-randomized design [27]. Briefly, RCTs were assessed according to criteria for adequacy of randomization method, adequacy of assignment concealment, similarity of treatment and

**Table 1. PICOS criteria.**

| Element | Focus | Further information |
|---|---|---|
| Patients | Individuals of any age with immunoglobulin A nephropathy (IgAN). Where available, information will be extracted separately for pre- and post-kidney failure (KF). | Exclude: Studies reporting a mixed population where data are not reported separately for IgAN patients meeting the inclusion criteria. |
| Interventions | Any intervention. | |
| Comparison | Any or no comparators. | |
| Outcomes | • Proteinuria variables (including, but not limited to, albuminuria [g/day, UACR], urine protein creatinine [PCR] ratio).<br>• Renal outcomes (including but not limited to, chronic kidney disease [CKD] progression, dialysis, transplant, KF, doubling of serum creatinine, 40% decrease in estimated glomerular filtration rate [eGFR], 57% decrease in eGFR, 50% decrease in eGFR, other survival or kidney failure criteria defined by study investigators).<br>• Cardiovascular events (including but not limited to nonfatal stroke, nonfatal myocardial infarction, cardiovascular death).<br>• Infections or episodes of sepsis.<br>• Creatinine clearance.<br>• Serum creatinine.<br>• Urinary protein excretion rate.<br>• Haematuria.<br>• Proportion of patients requiring immunosuppressive medication.<br>• Adverse events (AEs) (including overall rates, rates for severe and/or serious adverse events (SAEs) and treatment emergent adverse events, as well as rates for each specific AE).<br>• Drop-out rates due to treatment-related adverse events.<br>• Tolerability.<br>• Vital signs (including, but not limited to, systolic and diastolic blood pressure, pulse, temperature). | |
| Study designs | • Systematic reviews conducted in the most recent 5 years (for record checking only).<br>• Prospective clinical trials (Phases 1 to 4). | Exclude:<br>• Retrospective studies.<br>• Case reports.<br>• Non-systematic reviews. |
| Publication timeframe | Studies published from 1980 onwards. 1980 represents the earliest publication date of relevant studies. | |
| Geographic limitations | None. | |
| Language | English language abstract and full text. | |
| Databases to search | • PubMed.<br>• Ovid Embase.<br>• Cochrane Database of Systematic Reviews (Cochrane Library).<br>• Cochrane CENTRAL database (Cochrane Library).<br>• ClinicalTrials.gov.<br>• International Clinical Trials Registry Platform (ICTRP). | |
| Other search approaches | • Searches of the following conferences for the period 2019-2021:<br>• American Society of Nephrology (ASN).<br>• The European Renal Association – European Dialysis and Transplant Association (ERA-EDTA).<br>• International Society of Nephrology (ISN).<br>• UK Kidney Week.<br>• National Kidney Foundation meetings (NKF).<br>• Checking the reference lists of relevant systematic reviews published in the last 5 years.<br>• Checking the reference lists of included studies. | |
| Additional filters | Prioritization criteria:<br>• Patients: Total patient population ≥30 patients.<br>• Interventions: Pharmacological interventions.<br>• Outcomes: PCR, 24h urinary protein excretion rate (24h-PER) or eGFR.<br>• Study design: Randomized clinical trial. | Deprioritization criteria:<br>• Patients: Total patient population <30 patients.<br>• Interventions: Traditional or Chinese medicine, dietary interventions, physical activity, surgery.<br>• Study design: Non-randomized clinical trials, single-arm trials. |

control groups, adequacy of blinding methods, occurrence of unexpected imbalanced between groups due to drop-outs, evidence of authors withholding outcomes and inclusion of an intention to treat analysis and methods of tracking missing data [26]. While, non-randomized studies were assessed against three themes, study population selection, comparability of treatment groups and reporting of outcomes [27].

### Analysis

To enable comparison across studies, PCR data have been expressed as g/g (standardized from mg/g, etc.), and 24h-PER has been expressed as g/day (standardized from mg/day, etc.), unless otherwise stated. As the study designs and outcomes reported in the included studies were anticipated to be heterogeneous, a narrative synthesis was conducted to describe the findings. Narrative synthesis uses a textual approach (i.e., relies primarily on the use of words and text to summarize and explain the findings from different studies) [28]. This approach has been recommended for the synthesis of findings from multiple, heterogenous studies, when statistical meta-analysis or other forms of synthesis are not feasible [28].

## Results

### Search results

Database searches identified 6710 references, of which 1404 were duplicates and 4676 were excluded following title/abstract screening. From this, 630 full-text references were reviewed, 403 did not meet the PICOS criteria and were excluded. The 227 references identified for inclusion through electronic database searches and an additional 25 from supplementary searches resulted in a total of 254 references which reported outcomes from 183 studies (Fig 1). Following additional filtering, 154 references were excluded: 107 studies (25 references) focused on traditional Chinese medicines and dietary interventions and 81 references reported non-randomized trials, had small (<30 total patients) patient populations, investigated non-pharmacological treatments (surgery etc.) or did not report key clinical outcomes (PCR, 24h-PER or eGFR).

The remaining 100 references reported the results of 76 studies, which investigated the effect of pharmacological interventions in populations of at least 30 total patients and reported PCR, 24h-PER, or eGFR outcomes. These studies form the basis of this narrative synthesis.

### Summary of selected studies

Study design and population characteristics for each of the 76 studies selected for narrative synthesis are summarized in Table 2 (full study design description in S4 Table). Of the 76 studies selected for narrative synthesis, 8 were Phase 2 studies, 1 was Phase 2b, 1 was Phase 2/3, 7 were Phase 3, and 4 were Phase 4. The remaining 55 studies did not report a trial phase (Table 2). Fifty studies reported an open-label study design, 24 were double-blinded, and 2 did not describe blinding methods (Table 2). Criteria for entry into studies varied across the selected trials, including age, proteinuria measurements, and requirement for specific prior treatments. Study locations were reported in 73 of 76 selected studies with the highest number of studies conducted in China (24 studies) and Japan (14 studies, Table 2). Twelve conference abstracts [14,29-39] met the PICO criteria and were included in narrative synthesis but did not report study design and population characteristics for summary in S4 Table [14,29-39].

### Pharmacological therapies

The 76 studies selected for narrative synthesis were grouped by the primary intervention type investigated, inferred from the aims of each study (Fig 2). Eighteen studies investigated supportive therapies, including ACEi and ARBs either alone or in combination [9,40-51], 42 studies evaluated use of immunosuppressive or immunomodulatory therapies including steroids [12,14,16,17,29,30,32,35-37,39,52-97], and 8 studies investigated combination therapies, either in the same treatment group or different monotherapies across study treatment groups where the study aims focused on all treatments [18,22,98-103].

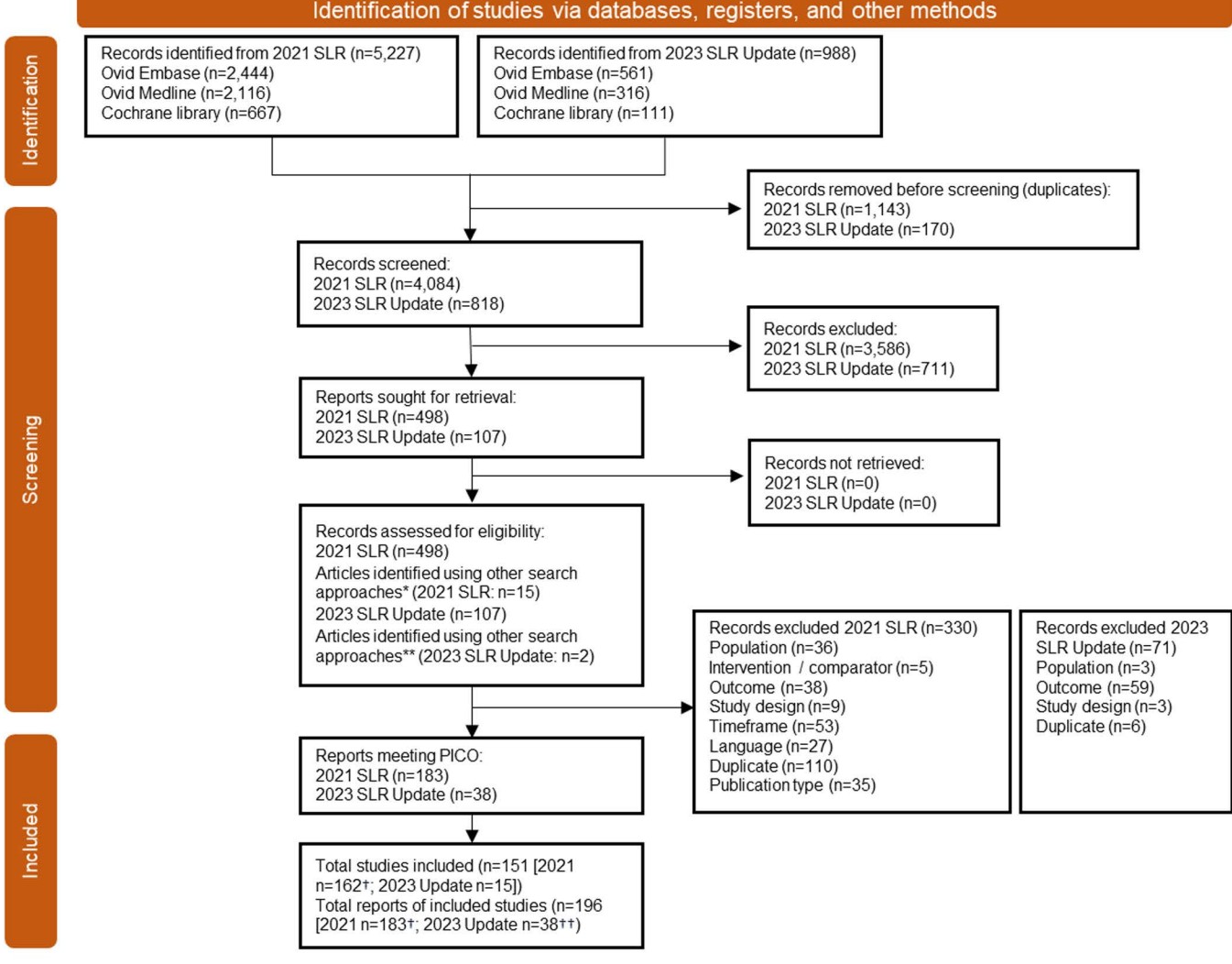

**Fig 1. PRISMA diagram summarizing reference screening.** * Patients: Total patient population ≥30 patients; Interventions: Pharmacological interventions; Outcomes: urinary protein creatinine ration, 24h-urinary protein excretion rate or estimated glomerular filtration rate; Study design: Randomized clinical trial.

An additional 7 studies investigated non-immunosuppressive therapies [104-114]. The STOP-IgAN trial, detailed in 5 references [12,30,31,55,56], and the 2023 NefIgArd study [17], evaluated the use of renin-angiotensin-system (RAS) inhibitors and immunosuppressive therapies. However, only STOP-IgAN reported outcomes stratified by RASi treatment, hence was described under both supportive therapies and immunosuppressive and immunomodulatory therapies.

## Supportive therapies

Eighteen studies investigated supportive therapy for IgAN and reported proteinuria outcomes; 4 studies reported PCR and 14 reported 24h-PER (Table 3), 7 studies reported eGFR (Table 4).

**Table 2. Study design summary of included studies.**

| Study attribute | Studies, N (%) |
|---|---|
| | N=76 |
| **Trial phase** | |
| Phase 2 | 8 (10.5%) |
| Phase 2/3 | 1 (1.3%) |
| Phase 2b | 1 (1.3%) |
| Phase 3 | 7 (9.2) |
| Phase 4 | 4 (5.2%) |
| Trial phase not reported | 55 (72.3%) |
| **Treatment concealment** | |
| Open-label | 50 (65.7%) |
| Double-blind | 24 (31.6%) |
| Not reported | 2 (2.6%) |
| **Study size** | |
| N = 30-50 | 28 (36.8%) |
| N = 51-100 | 28 (36.8%) |
| N = >100 | 20 (26.3%) |
| **Country** | |
| China | 24 (31%) |
| Japan | 14 (18.4%) |
| Europe | 10 (13.2%) |
| International | 10 (13.2%) |
| South Korea | 5 (6.6%) |
| Singapore | 4 (5.2%) |
| US | 3 (3.9%) |
| Hong Kong | 3 (3.9%) |
| Other | 3 (3.9%) |
| **Inclusion criteria** | |
| Age group | |
| Adults only (≥18 years) | 55 (72.3%) |
| Children only (<18 years) | 2 (2.6%) |
| Adults and children (≥10 years) | 15 (19.7%) |
| Not reported | 4 (5.2%) |
| **Follow-up time** | |
| 1-3 months | 3 (3.9%) |
| 4-6 months | 13 (17.1%) |
| 7-12 months | 11 (14.5%) |
| 13-24 months | 12 (15.8%) |
| >24 months | 34 (44.7%) |
| Not reported | 3 (3.9%) |

Shima et al. [40], Kohagura et al. [41], Jo et al. [42], and Park et al. [48] measured the impact of supportive therapies on PCR. Both valsartan for 6 months [42] and losartan for 3 months [48] resulted in significant reductions from base-line PCR following treatment (Table 3). Treatment with candesartan plus steroid pulse followed by oral prednisolone for 6 months and tonsillectomy within 6 months after steroid pulse therapy [41] resulted in a numerical reduction in PCR,

| Supportive therapy | Immunosuppressive & immunomodulatory therapies | Combination therapies | Non-immunosuppressive therapies |
|---|---|---|---|
| 1. ACEi<br>2. ARBs<br>3. ACEi + ARBs | 1. Atacicept<br>2. Azathioprine<br>3. Cemdisiran<br>4. Corticosteroids (including budesonide)<br>5. Cyclophosphamide<br>6. Fostamatinib<br>7. Hydroxychloroquine<br>8. Iptacopan<br>9. Leflunomide<br>10. Mizoribine<br>11. MMF<br>12. Ravulizumab<br>13. Rituximab<br>14. Sibeprenlimab<br>15. Tacrolimus<br>16. Telitacicept | 1. ACEi + Ticlopidine<br>2. ACEi + Urokinase<br>3. Anti-hyperlipidemic + ARB<br>4. ARB + Clopidogrel<br>5. ARB + Immunosuppression therapy<br>6. ARB + Mizoribine<br>7. Immunosuppression + anti-coagulant therapy | 1. Allopurinol<br>2. Dapagliflozin<br>3. Pioglitazone<br>4. Sodium cromoglycate<br>5. Sparsentan |

**Fig 2. IgAN treatment classes discussed in this review.** Abbreviations: ACEi, angiotensin -converting enzyme inhibitor; ARB, angiotensin II receptor blockers; MMF, mycophenolate mofetil.

although statistical significance of this change was not assessed. In total, 14 studies measured the impact of supportive therapy on 24h-PER, of which 10 reported a significant reduction in 24h-PER from baseline to follow-up (Table 3). Additionally, 2 references by Lennartz et al. [31,55] reported a significant increase in 24h-PER from baseline to 36-months following treatment with dual RAS blockade in the STOP-IgAN trial (Table 3). Five studies reported significantly lower 24h-PER in treatment groups compared with control groups at the final follow up (Table 3) [43,49,115-117].

Seven studies reported the effect of supportive therapy on eGFR [40,42-45,116,118,119]. Four studies reported eGFR remained stable (no significant change) from baseline to final follow-up (Table 4) [40,42,44,45,119]. Additionally, Hirai et al. [59] and Hou et al. [60] both reported no significant change in eGFR for any group during the study period, although this was only presented in figures, and as such, was not included in Table 4. Woo et al. [116] reported that losartan at 200 mg/day significantly slowed the rate of eGFR decline compared with losartan at 100 mg/day or enalapril at 10 or 20 mg/day (Table 4). Similarly, Li et al reported rate of eGFR decline as significantly slower for patients treated with valsartan than those given a placebo over 26 months (Table 4) [43]. An increase in eGFR was observed from baseline over 60 months after treatment with ramipril, although this change was not statistically significant [118].

Mortality and KF rates were low in the 3 studies that reported these outcomes, with the exception of patients in the control group of Woo et al. [117] where KF occurred in 21 of 38 patients (55%) compared with 7 of 37 patients (19%) in the treatment group (Table 5). Two studies reported overall adverse events (AE) leading to discontinuation. No patients discontinued treatment in Shima et al. [40] and 2 of 30 (6.6%) patients in the ramipril group discontinued treatment due to AEs in Li et al. [118](Table 6).

## Immunosuppressive/immunomodulatory therapies

Thirty-eight studies of immunosuppressive/immunomodulatory therapies reported proteinuria outcomes (13 reported PCR and 30 reported 24h-PER; Table 7), 22 reported eGFR outcomes (Table 8), 18 reported mortality and KF rates (Table 5)

Table 3. Proteinuria outcomes in patients treated with supportive therapy.

| Author/ Trial name/ NCT | Treatment (N) | FU, mo[a] | Age, yrs[b] | Baseline SCr, mg/dL[b] | Baseline Proteinuria[bc] | FU Proteinuria[abc] | Change from baseline[d] | p-value |
|---|---|---|---|---|---|---|---|---|
| Lennartz et al. [31], Lennartz et al. [55] & Lennartz et al. [30]; STOP-IgAN/ NCT00554502 | Single RAS blockade (N=91/N=89) | 36 | NR | NR | NR | NR | -0.3g/g | • p=significant from baseline to FU (Dual) |
| | Dual RAS blockade (N=34) | 36 | NR | NR | NR | NR | 0.25g/day | |
| | Single-dual RAS blockade (N=30) | 36 | NR | NR | NR | NR | -0.32g/day | |
| | Single RAS blockade (N=82) | 36 | 45.5 (12.2) | NR | NR | NR | -0.3g/day | • p=0.011 between treatment arms |
| | Dual RAS blockade (N=30) | 36 | 44.5 (12.8) | NR | NR | NR | 0.1g/g | |
| Shima et al. [40]; JSKDC01/ C000000006 | Lisinopril [0.1mg/kg] (N=31) | 24 | NR | 0.5 (0.1) | 0.7 (0.8) g/g | 0.3 (0.3) g/g | NR | • p=0.621 |
| | Lisinopril [0.1mg/kg] + losartan [0.7mg/kg] (N=31) | 24 | NR | 0.5 (0.1) | 0.5 (0.3) g/g | 0.2 (0.2) g/g | NR | |
| Kohagura et al. [41]; ACTRN12610000516088 | Standard therapy (steroid pulse, prednisolone and tonsillectomy) (N=37) | 1.5 | 35.8 (14.6) | 0.8 (0.2) | 1.02 (NR) g/g | 0.21 (NR) g/g | NR | • NR |
| | Standard therapy (steroid pulse, prednisolone and tonsillectomy) + candesartan [2-8mg/day] (N=40) | 1.5 | 36.3 (12.8) | 0.8 (0.2) | 0.97 (NR) g/g | 0.11 (NR) g/g | NR | |
| Jo et al. [42] | Valsartan [40 mg] (N=23) | 6 | 37.9 (10.8) | 0.9 (0.2) | 0.57 (0.18) g/g | 0.42 (0.28) g/g | NR | • p=0.015 valsartan 40mg baseline vs follow-up • p<0.001 valsartan 80mg baseline vs follow-up • p=0.599 40mg vs 80mg valsartan |
| | Valsartan [80 mg](N=20) | 6 | 39.9 (13.8) | 0.9 (0.1) | 0.68 (0.24) g/g | 0.38 (0.19) g/g | NR | |
| Woo et al. [116] | Losartan [200 mg/day] (N=63) | 74[e] | 34 (10) | NR | 2.2 (0.9) g/day | 1.2 (0.8) g/day | NR | • p<0.0005 between treatment arms |
| | Losartan [100mg/day] (N=43) | 74[e] | 32 (12) | NR | 2.0 (0.9) g/day | 1.6 (0.9) g/day | NR | |
| | Enalapril [20 mg/day] (N=61) | 74[e] | 32 (10) | NR | 2.2 (1.6) g/day | 1.7 (1.0) g/day | NR | |
| | Enalapril [10 mg/day] (N=40) | 74[e] | 34 (11) | NR | 2.3 (1.5) g/day | 1.7 (0.9) g/day | NR | |

(Continued)

| Author/ Trial name/ NCT | Treatment (N) | FU, mo[a] | Age, yrs[b] | Baseline SCr, mg/dL[b] | Baseline Proteinuria[bc] | FU Proteinuria[abc] | Change from baseline[d] | p-value |
|---|---|---|---|---|---|---|---|---|
| Shimizu et al. [119] | Losartan [12.5mg/day] (N=18) | 12 | 36.0 (8.5) | 1.0 (0.2) | 0.81 (0.52) g/day | **0.39 (0.42) g/day** | NR | • p=0.006 from baseline to FU (losartan)<br>• p=NS from baseline to FU (antiplatelet) |
| | Antiplatelet (N=18) | 12 | 35.7 (8.1) | 0.9 (0.2) | 0.73 (0.36) g/day | **0.66 (0.41) g/day** | NR | |
| Coppo et al. [8]; IGACE | Benazepril [0.2mg/kg] (N=32) | 38 (NR-58)[e] | 21.8 (6.3) | NR | 1.61 (0.70) g/day | **0.94 (0.98) g/day** | NR | • p=0.002 from baseline to FU (benazepril)<br>• p=NS from baseline to FU (placebo) |
| | Placebo (N=34) | 38 (NR-58)[e] | 19.3 (6.1) | NR | 1.87 (0.74) g/day | 1.80 (1.34) g/day | NR | |
| Horita et al. [115] | Prednisolone [5-30mg/dL] + losartan [50mg/day] (N=22) | 25 | 33.5 (12) | 0.8 (0.2) | 1.6 (0.6) g/day | **0.3 (0.1) g/day** | NR | • p<0.01 from baseline to FU (prednisolone+ losartan)<br>• p<0.05 from baseline to FU (prednisolone)<br>• p<0.05 between treatment arms at FU |
| | Prednisolone [5-30mg/dL] (N=18) | 25 | 32.3 (10.6) | 0.7 (0.1) | 1.4 (0.4) g/day | **0.5 (0.1) g/day** | NR | |
| Woo et al. [117] | Enalapril [5-10mg/day] and/ or losartan [50-100mg/day] (N=37) | 60 | 36 (11) | 1.6 (0.4) | 2.1 (0.8) g/day | 1.1 (0.9) g/day | NR | • p<0.002 between treatment arms at FU |
| | Control (N=38) | 60 | 34 (11) | 1.5 (0.4) | 2.3 (1.6) g/day | 1.9 (1.0) g/day | NR | |
| Horita et al. [45] & Horita et al. [44] | Temocapril [1mg/day] (N=14) | 12 | 43 (10) | 0.84 (0.17) | 0.60 (0.21) g/day[f] | **0.30 (0.23) g/day[f]** | -47.9% (32.8)[f] | • p<0.05 from baseline to FU (temocapril)<br>• p<0.05 from baseline to FU (losartan)<br>• p<0.05 from baseline to FU (temocapril + losartan) |
| | Losartan [12.5mg/day] (N=16) | 12 | 42 (9) | 0.89 (0.14) | 0.83 (0.44) g/day[f] | **0.44 (0.25) g/day[f]** | -40.9% (52.2)[f] | |
| | Temocapril [1mg/day] + losartan [12.5mg/day] (N=13) | 12 | 38 (9) | 0.84 (0.2) | 0.80 (0.33) g/day[f] | **0.23 (0.17) g/day[f]** | -72.2% (15.7)[f] | |
| | Temocapril [1mg/day] (N=10) | 6 | 39.6 (10.8) | 0.85 (0.21) | 0.73 (0.36) g/day[f] | **0.44 (0.31) g/day[f]** | -41.3% | • p<0.05 from baseline to FU (temocapril)<br>• p<0.05 from baseline to FU (losartan)<br>• p<0.05 from baseline to FU (temocapril + losartan) |
| | Losartan [12.5mg/day] (N=10) | 6 | 42.7 (12.0) | 0.88 (0.17) | 0.81 (0.44) g/day[f] | **0.55 (0.38) g/day[f]** | -36.6% | |
| | Temocapril [1mg/day] + losartan [12.5mg/day] (N=11) | 6 | 39.6 (10.4) | 0.83 (0.19) | 0.75 (0.30) g/day[f] | **0.28 (0.20) g/day[f]** | -63.2% | |

*(Continued)*

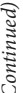

**Table 3.** (Continued)

| Author/ Trial name/ NCT | Treatment (N) | FU, mo[a] | Age, yrs[b] | Baseline SCr, mg/dL[b] | Baseline Proteinuria[bc] | FU Proteinuria[bc] | Change from baseline[d] | p-value |
|---|---|---|---|---|---|---|---|---|
| Li et al. [43]; HKVIN | Valsartan [80mg/day] (N=54) | 26 | 40 (10) | 1.11 (0.48) | 1.8 (1.2) g/day | 1.23 (1.25) g/day | -33.5 (40.8)% [-0.66 (0.89) g/day] | • p<0.001 from baseline to FU (valsartan)<br>• p=0.16 from baseline to FU (placebo)<br>• p=0.001 between arms |
| | Placebo (N=55) | 26 | 41 (9) | 1.29 (0.54) | 2.3 (1.7) g/day | 1.97 (1.67) g/day | 15 (67.2)% [0.08 (1.48) g/day] | |
| Kanno et al. [46] | Temocapril [1-2mg/day] or trandolapril [mg/day] (N=26) | 36 | 35 (2) | 1.07 (0.13) | 1.09 (0.16) g/day | **0.79 (0.36) g/day** | NR | • p<0.05 from baseline to FU (temocapril or trandolapril)<br>• p=NS from baseline to FU (amlodipine) |
| | Amlodipine [2.5-5mg/day] (N=23) | 36 | 35 (3) | 1.02 (0.08) | 1.10 (0.15) g/day | 1.33 (0.62) g/day | NR | |
| Kim et al. [47] | Ramipril [5mg/day] (N=19, IgAN only) | 3 | 30 (1) | NR | 4.0 (0.4) g/day[f] | 4.2 (0.3) g/day[f] | NR | • NR |
| | Ramipril [5mg/day] + candesartan [4mg/day] (N=19, IgAN only) | 3 | | | | 3.1 (0.3) g/day[f] | NR | |
| | Placebo (N=19, IgAN only) | 3 | | | | 4.3 (0.2) g/day[f] | NR | |
| Park et al. [48] | Control (N=22) | 3 | 28.2 (8.9) | NR | NR | NR | NR | • p<0.05 baseline to FU (losartan)<br>• p=NS baseline to FU (amlodipine) |
| | Losartan [50 mg/day](N=20) | 3 | 39.3 (8.7) | 1.5 (0.6) | 1.9 (0.8) g/g | 1.1 (1.2) g/g | NR | |
| | Amlodipine [5mg/day] (N=16) | 3 | 44.3 (13.4) | 1.5 (0.6) | 1.7 (1.1) g/g | 2.0 (1.4) g/g | NR | |
| Praga et al. [49] | Enalapril [5-40 mg/day] (N=23) | 78[e] | 27.8 (12) | 1 (0.2) | 2.0 (1.3) g/day | **0.9 (1.0) g/day** | NR | • p<0.001 from baseline to FU (enalapril)<br>• p=NS from baseline to FU (control)<br>• p<0.001 between treatment arms |
| | Control (N=21) | 74[e] | 29.9 (12.3) | 0.9 (0.2) | 1.7 (0.8) g/day | 2.0 (1.8) g/day | NR | |

*(Continued)*

**Table 3.** (Continued)

| Author/ Trial name/ NCT | Treatment (N) | FU, mo[a] | Age, yrs[b] | Baseline SCr, mg/dL[b] | Baseline Proteinuria[bc] | FU Proteinuria[abc] | Change from baseline[d] | p-value |
|---|---|---|---|---|---|---|---|---|
| Woo et al. [9] | Enalapril [5 mg/day] and/or losartan [50 mg/day] (N=21) | 94 (22)[b] | 39 (10) | 2.0 (0.8) | 2.2 (1.2) g/day | 1.8 (1.6) g/day | NR | • p<0.05 enalapril/ losartan vs control at FU<br>• p=NS baseline to FU (enalapril/ losartan) |
| | Control (N=20) | 86 (22)[b] | 37 (6) | 1.8 (0.8) | 2.1 (1.1) g/day | 2.9 (1.8) g/day | NR | • p<0.002 baseline to FU (control)<br>• p<0.005 responders vs non-responders at FU |
| | Enalapril [5 mg/day] and/or losartan [50 mg/day] -responders (N=10) | 104 (9)[b] | 38 (7) | 1.7 (0.6) | 2.3 (1.1) g/day | **0.7 (0.5) g/day** | NR | • p<0.002 baseline to FU (responders) |
| | Enalapril [5 mg/day] and/or losartan [50 mg/day] - non-responders (N=11) | 84 (25)[b] | 41 (13) | 2.3 (0.9) | 2.1 (1.4) g/day | 2.8 (1.7) g/day | NR | |
| Nakamura et al. [50] | Verapamil [120 mg/day] (N=8) | 3 | NR | 0.9 (0.2) | 1.8 (0.6) g/day | 1.4 (0.5) g/day | NR | • p<0.05 from baseline to FU (verapamil)<br>• p<0.01 from baseline to FU (trandolapril) |
| | Trandolapril [2 mg/day] (N=8) | 3 | NR | 0.8 (0.2) | 1.9 (0.7) g/day | 1.2 (0.5) g/day | NR | • p<0.01 from baseline to FU (candesartan cilexetil) |
| | Candesartan [8 mg/day] (N=8) | 3 | NR | 0.7 (0.2) | 1.8 (0.8) g/day | 1.1 (0.6) g/day | NR | |
| | Placebo (N=8) | 3 | NR | 0.8 (0.2) | 1.1 (0.6) g/day | 1.7 (0.7) g/day | NR | |
| Maschio et al. [51] | Fosinopril [20 mg/day] (N=39) | 4 | 33.2 (11.4) | 1.0 (0.2) µmol/L | 1.74 (0.84) g/day | 1.37 (0.98) g/day | NR | • p=0.017 from baseline to FU (fosinopril)<br>• p=NS from baseline to FU (placebo) |
| | Placebo (N=39) | 4 | | | | 1.79 (1.20) g/day | NR | |

Abbreviations: FU, follow-up; IgAN, IgA nephropathy; IQR, interquartile range; mo, months; NR, not reported; NS, not significant; PCR, urine protein creatinine ratio; RAS, renin-angiotensin-system; SCr, serum creatinine; SD, standard deviation; SEM, standard error of the mean; yrs, years.

[a]Follow-up durations are presented in months and have been calculated into months (4 weeks/ month; 12 months/ year). Follow-up refers to the final and longest duration of time reported or the duration at which the authors presented the change from baseline. The percentage change from baseline is presented for this follow-up duration; [b]Presented as mean (SD) unless otherwise stated; [c]Proteinuria was reported as PCR or 24h-PER and are indicated here with units g/g or g/day, respectively; [d]Change from baseline to last follow-up as mean (SD) unless otherwise stated, units are given with values; [e]Presented as median (range); [f]Presented as mean (SEM).

**Bold** indicates clinically significant 24h-PER (<1.0g/day) at follow-up.

**Table 4. eGFR outcomes in patients treated with supportive therapies.**

| Author/ Trial name/ NCT | Treatment (N) | FU[a], mo | Age[b], yrs | Baseline SCr[b], mg/dL | Baseline eGFR[b], ml/min/1.73m² | FU eGFR[b], ml/min/1.73m² | Change in eGFR[bc] | p-value |
|---|---|---|---|---|---|---|---|---|
| Shima et al. [40]; JSKDC01/C000000006 | Lisinopril [0.1 mg/kg] (N=31) | 24 | NR | 0.5 (0.1) | 120.1 (13.8) | 124.3 (14.0) | NR | • p=0.900 for difference of changes between the groups |
| | Lisinopril [0.1 mg/kg] + losartan [0.7 mg/kg] (N=31) | 24 | NR | 0.5 (0.1) | 121.3 (15.3) | 124.9 (19.4) | NR | |
| Jo et al. [42] | Valsartan [40 mg] (N=23) | 6 | 37.9 (10.8) | 0.9 (0.2) | 107.7 (26.7) | 100.1 (38.0) | NR | • p=0.946 between treatment groups at FU |
| | Valsartan [80 mg] (N=20) | 6 | 39.9 (13.8) | 0.9 (0.1) | 96.7 (19.8) | 99.2 (32.5) | NR | |
| Li et al. [118]; NCT01225445 | Ramipril [2.5 mg/day] (N=30) | 60 | 42.2 (11) | 77.4 (16.8) µmol/L | 106.8 (20.9) | 108.1 (29.0) | -0.39 (2.57)[d] | • p=0.4 ramipril baseline v no treatment baseline |
| | No treatment (N=30) | 60 | 41 (7.5) | 75.4 (12.2) µmol/L | 102.9 (15.2) | 105.7 (17.7) | -0.59 (1.63)[d] | • p=0.7 rate of decline |
| Woo et al. [116] | Losartan [200mg/day] (N=63) | 72 | 34 (10) | NR | 63.5 (24.2)[f] | 59.1 (31.8)[f] | Slope: -0.75 (3.1)[e] | • p=0.452 between groups at baseline |
| | Losartan [100mg/day] (N=43) | 72 | 32 (12) | NR | 61.2 (18.4)[f] | 40.2 (27.6)[f] | Slope: -3.5 (3.2)[e] | • p<0.0005 between groups at FU · • p<0.0005 between groups in decrease in eGFR |
| | Enalapril [20 mg/day] (N=61) | 72 | 32 (10) | NR | 62.0 (20.8)[f] | 41.3 (27.9)[f] | Slope: -3.5 (3.3)[e] | |
| | Enalapril [10 mg/day] (N=40) | 72 | 34 (11) | NR | 60.9 (19.8)[f] | 42.3 (26.6)[f] | Slope: -3.2 (2.6)[e] | |
| Shimizu et al. [119] | Losartan [12.5mg/day](N=18) | 12 | 36.0 (8.5) | 1 (0.2) | 72.0 (15.9) | 71.8 (17.1) | NR | • NR |
| | Antiplatelet (N=18) | 12 | 35.7 (8.1) | 0.9 (0.2) | 75.4 (18.1) | 76.1 (17.3) | NR | |
| Horita et al. [45] & Horita et al. [44] | Temocapril [1 mg/day] (N=14) | 12 | 43 (10) | 0.84 (0.17) | 89.8 (22.9)[g] | 80.4 (26.5)[g] | NR | • p=NS from baseline to FU for all treatment arms |
| | Losartan [12.5mg/day] (N=16) | 12 | 42 (9) | 0.89 (0.14) | 88.0 (18.2)[g] | 87.0 (24.7)[g] | NR | |
| | Temocapril [1 mg/day] + losartan [12.5mg/day] (N=13) | 12 | 38 (9) | 0.84 (0.2) | 95.3 (22.4)[g] | 89.2 (25.8)[g] | NR | |
| | Temocapril [1 mg/day] (N=10) | 6 | 39.6 (10.8) | 0.85 (0.21) | 92.5 (17.2)[g] | 87.5 (17.9)[g] | NR | • p=NS from baseline to FU for all treatment arms |
| | Losartan [12.5mg/day] (N=10) | 6 | 42.7 (12.0) | 0.88 (0.17) | 88.3 (19.8)[g] | 85.8 (20.7)[g] | NR | |
| | Temocapril [1 mg/day] + losartan [12.5mg/day] (N=11) | 6 | 39.6 (10.4) | 0.83 (0.19) | 91.5 (24.6)[g] | 82.7 (29.1)[g] | NR | |
| Li et al. [43]; HKVIN | Valsartan [80mg/day] (N=54) | 26 | 40 (10) | 1.11 (0.48) | 85.90 (37.30) | 72.36 (34.20) | Slope: -5.62 (6.79)[e] | • p=0.025 unadjusted difference in rates of decrease, valsartan vs placebo |
| | Placebo (N=55) | 26 | 41 (9) | 1.29 (0.54) | 72.47 (35.16) | 63.39 (34.79) | Slope: -6.89 (6.17)[e] | |

**Abbreviations:** eGFR, estimated glomerular filtration rate; FU, follow-up; IQR, interquartile range; mo, months; NR, not reported; NS, not significant; SCr, serum creatinine; SD, standard deviation; SEM, standard error of the mean; yrs, years.

[a]Follow-up durations are presented in months and have been calculated into months (4 weeks/ month; 12 months/ year). Follow-up refers to the final and longest duration of time reported or the duration at which the authors presented the change from baseline. The percentage change from baseline is presented for this follow-up duration; [b]Data is presented as mean (SD) unless otherwise stated; [c]Change from baseline is reported in ml/min/1.73m² unless stated as percentage; [d]Presented as ml/min/1.73 m²/year; [e]Presented as ml/min/year; [f]Presented as ml/min; [g]Presented as mean (SEM).

Table 5. Mortality and KF rates in patients with IgAN during clinical trials.

| Author | Treatment (N) | FU, mo | Age[a], yrs | KF rate | Mortality rate |
|---|---|---|---|---|---|
| Mathur et al. [97]; ENVISION/ NCT04287985 | Sibeprenlimab [2 mg/kg] (N=38) | 16[g] | 41 (25-71)[b] | NR | 0/38 |
| | Sibeprenlimab [4 mg/kg] (N=41) | 16[g] | 39 (20-73)[b] | NR | 0/41 |
| | Sibeprenlimab [8 mg/kg] (N=38) | 16[g] | 42 (23-72)[b] | NR | 0/38 |
| | Placebo (N=38) | 16[g] | 36 (18-52)[b] | NR | 1/38 |
| Barratt et al. [16] & Lafayette et al. [17][d]; NefIgArd/ NCT03643965[15] | TRF-B (N=182)[d] | 24 | 43 (36-50)[b] | NR | 1/182 |
| | Placebo (N=182)[d] | 24 | 42 (34-49)[b] | NR | 1/182 |
| Heerspink et al. [22][e] & Rovin et al. [18][f]; PRO-TECT/ NCT03762850 | Sparsentan (N=202) | 28[f] | 46.6 (12.8) | 9/202 | 0% (0/202) |
| | Irbesartan (N=202) | 28[f] | 45.4 (12.1) | 11/202 | <1% (1/202) |
| Hou et al. [89]; MAIN/ NCT01854814 | MMF (N=85) | 36 | 35 (8.7) | 3/85 | 0/85 |
| | Supportive care (N=85) | 36 | 38.2 (9.8) | 7/85 | 1/85 |
| Kim et al. [38][g] & Lv et al. [63][h]; TESTING/ NCT01560052 | MP (N=121)[g]/ (N=257)[h] | 7[h] | 35.6 (29.4-46.3)[b] | NR | 2.3% (6/257) |
| | Placebo (N=120)[g]/ (N=246)[h] | 7[h] | 36.6 (29-45.9)[b] | NR | 1.2% (3/246) |
| Tam et al. [96]; NCT02112838 | Placebo (N=25) | 6 | 40 (20-59)[i] | NR | 0/25 |
| | Fostamatinib [100 mg] (N=26) | 6 | 42 (19-67)[i] | NR | 0/26 |
| | Fostamatinib [150 mg] (N=25) | 6 | 41 (20-68)[i] | NR | 1/25 |
| www.clinicaltrials.gov[93]; (NCT03841448) | Cemdisiran (N=22) | 88[j] | 40.5 (10.1) | NR | 1/22 |
| | Placebo (N=9) | 88[j] | 37.6 (10.4) | NR | 1/9 |
| Han et al. [52]; NCT02981212 | MMF + corticosteroid (N=24) | 48 | 44.0 (10.6) | NR | 1/24 |
| | Control (N=20) | 48 | 46.1 (7.8) | NR | 0/20 |
| Wheeler et al. [98]; DAPA-CKD/ NCT03036150 | Dapagliflozin (N=137) | 36 | 52.2 (13.1) | 5/137 | 0/137 |
| | Placebo (N=133) | 36 | 50.1 (13.1) | 16/133 | 0/133 |
| Lennartz et al. [55] & Rauen et al. [12] & Rauen et al. [56]; STOP-IgAN/ NCT00554502 | Single RAS blockade | 36 | 45.5 (12.2) | 1/82 | NR |
| | Dual RAS blockade | 36 | 44.5 (12.8) | 0/30 | NR |
| | High eGFR[k] – supportive therapy (RAS block-ade) (N=54) | 45.6 (11.9)[a] | 1.4 (0.5) | 5/55 | 1/54 |
| | High eGFR[k] – supportive therapy (RAS blockade) + immunosuppression (N=55) | 41.7 (13.3)[a] | 1.3 (0.4) | 1/26 | 0/55 |
| | Low eGFR[k] – supportive therapy (RAS blockade) (N=26) | 46.0 (14.0)[a] | 2.0 (0.6) | 1/27 | 0/26 |
| | Low eGFR[k] – supportive therapy (RAS blockade) + immunosuppression (N=27) | 45.1 (12.8)[a] | 2.2 (0.7) | 5/27 | 1/27 |
| | Supportive therapy (RAS blockade) (N=80) | 45.8 (12.5)[a] | 1.6 (0.6) | 6/80 | 1/80 |
| | Supportive therapy (RAS blockade) + immuno-suppression (N=82) | 42.8 (13.1)[a] | 1.6 (0.7) | 6/81 | 1/82 |
| Lv et al. [62]; TESTING/ NCT01560052 | MP (N=136) | 36 | 38.6 (11.5) | NR | 2/134 |
| | Placebo (N=126) | 36 | 38.6 (10.7) | NR | 1/126 |

(Continued)

Table 5. (Continued)

| Author | Treatment (N) | FU, mo | Age[a], yrs | KF rate | Mortality rate |
|---|---|---|---|---|---|
| Min et al. [64] | Prednisone (N=45) | 60 | 3.6 (11.5) | 5/45 | 0/45 |
| | Leflunomide + prednisone (N=40) | 60 | 36.9 (10.5) | 3/40 | 0/40 |
| Kamei et al. [108] | Prednisolone + azathioprine + heparin-warfarin + dipyridamole (N=40) | 24 | 12.2 (3) | 2/40 | 0/40 |
| | Heparin-warfarin + dipyridamole (N=38) | 24 | 11.6 (2.3) | 5/38 | 0/38 |
| Pozzi et al. [120]; NCT00755859/ NCT01392833 | MP + prednisone + azathioprine (N=101) | 58.8 (36, 16.8)[b] | 34.8 (27.7, 43.9)[b] | NR | 1/101 |
| | MP + prednisone (N=106) | 58.8 (36, 16.8)[b] | 40.5 (30.3, 51.3)[b] | NR | 3/106 |
| Tang et al. [73] & Tang et al. [72]; NCT00863252 | MMF (N=20) | 18 | 42.1 (2.6) | 2/20 | 0/20 |
| | Conventional therapy (N=20) | 18 | 43.3 (2.8) | 9/20 | 0/20 |
| Manno et al. [75] | Ramipril (N=49) | 60 | 34.9 (11.2) | 7/48 | 0/49 |
| | Ramipril + prednisone (N=48) | 60 | 31.8 (11.3) | 1/48 | 0/48 |
| Coppo et al. [8] | Benazepril | 38 (NR-58)[b] | 21.8 (6.3) | NR | 1/32 |
| | Placebo | 38 (NR-58)[b] | 19.3 (6.1) | NR | 0/32 |
| Woo et al. [117] | Enalapril and/or losartan (N=37) | 60 | 36 (11) | 7/37 | 0/37 |
| | Control (N=38) | 60 | 34 (11) | 21/38 | 0/38 |
| Frisch et al. [78] | MMF (N=17) | 14.75 (NR)[a] | 39 (19, 72)[l] | 5/17 | 0/17 |
| | Placebo (N=15) | 18 (NR)[a] | 37 (22, 59)[l] | 2/15 | 0/15 |
| Katafuchi et al. [82] | Prednisolone (N=43) | 60 | 33.6 (13.4) | 3/43 | 0/43 |
| | Control (N=47) | 60 | 32.5 (10.8) | 3/47 | 0/47 |
| Ballardie and Roberts[84] | Prednisolone + cyclophosphamide + azathioprine (N=19) | 48 | >45 | NR | 1/19 |
| | Control, no immunosuppression (N=19) | 36 | | NR | 0/19 |
| Locatelli et al. [85] | MP + prednisone (N=43) | 72 | NR | NR | 0/43 |
| | Supportive treatment (diuretics, antihypertensive drugs and antiplatelet agents) (N=43) | 72 | NR | 2/43 | 0/43 |
| Woo et al. [113] | Cyclophosphamide + dipyridamole + warfarin (N=27) | 94 (22)[a] | 25 (6) | 6/27 | 0/27 |
| | Control (N=21) | 86 (22)[a] | 26 (9) | 7/21 | 0/21 |
| | Continuation of dipyridamole + warfarin (N=13) | 104 (9)[a] | 25 (6) | 0/13 | 0/13 |
| | Control continuation (N=14) | 84 (25)[a] | 26 (9) | 6/14 | 0/14 |

**Abbreviations:** eGFR, glomerular filtration rate; FU, follow-up; IgAN, IgA nephropathy; KF, End-stage kidney disease; MMF, mycophenolate mofetil; mo, months; MP, methyl-prednisone; NR, not reported; RAS, renin-angiotensin system; yrs, years.

[a]Presented as mean (SD); [b]Presented as median (IQR); [c,d]Denotes which reference in the study reported which value; [e,f]Denotes which reference in the study reported which value; [g,h]Denotes which reference in the study reported which value; [i]Mean (range); [j]Maximum; [k]Patients in the high-eGFR arm had eGFR ≥60ml/min/1.73m² at baseline and patients in the low eGFR arm had eGFR between 30 and 59ml/min/1.73m² at baseline; [l]Presented as median (range).

**Table 6. Safety outcomes.**

| Treatment (N) | AEs, N (%) | |
|---|---|---|
| Supportive therapies | | |
| C000000006 – Shima et al. [40] | | |
| Lisinopril (N= 31) | • Discontinuation due to AE: 0 (0%) | |
| Lisinopril + losartan (N=31) | | |
| NCT0122545 – Li et al. [118] | | |
| Ramipril (N=30) | • AEs leading to treatment discontinuation: 2 (6.6%) | |
| No treatment (N=30) | • NR | |
| Immunosuppressants/ immunomodulatory therapies | | |
| NCT03373461 [87][a], [135] [135][b] | | |
| Iptacopan [10mg] (N=20) | • Any AE: 14 (70%)[a]<br>• SAE: 0 (0%)[a,b]<br>• Any TEAE: 14 (70%)[b]<br>• Mild TEAEs: 12 (60%)[b]<br>• Moderate TEAEs: 4 (20%)[b]<br>• Severe TEAEs: 0 (0%)[b] | • AEs leading to discontinuation: 0 (0%)[b]<br>• TRAEs: 5 (25%)[b] |
| Iptacopan [50mg] (N=19) | • Any AE: 16 (84.2%)[a]<br>• SAE: 1 (5.3%)[a,b]<br>• Any TEAE: 16 (84.2%)[b]<br>• Mild TEAEs: 15 (78.9%)[b]<br>• Moderate TEAEs: 2 (10.5%)[b]<br>• Severe TEAEs: 1 (5.3%)[b] | • AEs leading to discontinuation: 1 (5.3%)[b]<br>• TRAEs: 5 (26.3%)[b] |
| Iptacopan [100mg] (N=22) | • Any AE: 15 (68.2%)[a]<br>• SAE: 0 (0%)[a,b]<br>• Any TEAE: 15 (68.2%)[b]<br>• Mild TEAEs: 15 (68.2%)[b]<br>• Moderate TEAEs: 2 (9.1%)[b]<br>• Severe TEAEs: 0 (0%)[b] | • AEs leading to discontinuation: 0 (0%)[b]<br>• TRAEs: 7 (31.8%)[b] |
| Iptacopan [200mg] (N=26) | • Any AE: 14 (53.9%)[a]<br>• SAE: 0 (0%)[a,b]<br>• Any TEAE: 14 (53.8%)[b]<br>• Mild TEAEs: 13 (50%)[b]<br>• Moderate TEAEs: 3 (11.5%)[b]<br>• Severe TEAEs: 0 (0%)[b] | • AEs leading to discontinuation: 0 (0%)[b]<br>• TRAEs: 2 (7.7%)[b] |
| Placebo (N=25) | • Any AE: 17 (68%)[b]<br>• SAE: 1 (4%)[a,b]<br>• Any TRAE: 18 (72%)[b]<br>• Mild TRAEs: 17 (68%)[b]<br>• Moderate TRAEs: 7 (28%)[b]<br>• Severe TRAEs: 0 (0%)[b] | • AEs leading to discontinuation: 2 (8%)[b]<br>• TRAEs: 5 (20%)[b] |
| NCT04564339 [33] | | |
| Ravulizumab (N=43) | • Any AE: 32 (74.4%)<br>• Any SAE: 1 (2.3) | • TRAE: 9 (20.9%) |
| Placebo (N=23) | • Any AE: 19 (82.6%)<br>• Any SAE: 0 (0%) | • TRAE: 6 (26.1%) |
| NefIgArd/ NCT03643965 - Barratt et al. [16][c] & Lafayette et al. [17][d] | | |
| TRF-B (N=97)[c] | • Any TEAE: 84 (86.6%)<br>• Mild TEAE: 49 (50.5%)<br>• Moderate TEAE: 31 (32%)<br>• Severe TEAE: 4 (4.1%)<br>• Serious TEAE: 11 (11.3%) | • Infection AEs: 38 (39.2%)<br>• Treatment discontinuation: 9 (9.3%)<br>• AEs leading to death: 0 (0%) |

*(Continued)*

| Treatment (N) | AEs, N (%) | |
|---|---|---|
| Placebo (N=102)[c] | • Any TEAE: 73 (73%)<br>• Mild TEAE: 46 (46%)<br>• Moderate TEAE 26 (26%)<br>• Severe TEAE: 1 (1%)<br>• Serious TEAE: 5 (5%) | • Infection AEs: 41 (41%)<br>• Treatment discontinuation: 1 (1%)<br>• AEs leading to death: 0 (0%) |
| TRF-B (N=182)[d] | • Any TEAE<br>Treatment period:159 (87%)<br>Post-treatment period: 127 (73%)<br>• Mild TEAE<br>Treatment period: 93 (51%)<br>Post-treatment period: 62 (35%)<br>• Moderate TEAE<br>Treatment period: 57 (31%)<br>Post-treatment period: 49 (28%) | • Severe TEAE<br>Treatment period: 9 (5%)<br>Post-treatment period: 16 (9%)<br>• Serious TEAE<br>Treatment period: 18 (10%)<br>• Post-treatment period: 14 (8%)<br>• Treatment-related TEAEs: 4 (2%)<br>• Infection AEs: 63 (35%)<br>• TEAEs leading to death: 1 (1%)<br>• TEAEs leading to discontinuation: 17 (9%) |
| Placebo (N=182)[d] | • Any TEAE<br>Treatment period:15 (69%)<br>Post-treatment period: 124 (71%)<br>• Mild TEAE<br>Treatment period: 75 (41%)<br>Post-treatment period: 73 (42%)<br>• Moderate TEAE<br>Treatment period: 46 (25%)<br>Post-treatment period: 43 (25%) | • Severe TEAE<br>Treatment period: 4 (2%)<br>• Post-treatment period: 8 (5%)<br>• Serious TEAE<br>Treatment period: 9 (5%)<br>• Post-treatment period: 14 (8%)<br>• Treatment-related TEAEs: 4 (2%)<br>• Infection AEs: 57 (31%)<br>• TEAEs leading to death: 1 (1%)<br>• TEAEs leading to discontinuation: 3 (2%) |
| MAIN/ NCT01854814 - Hou et al. [89] | | |
| MMF (N=85) | • SAEs: 4 (NR) | • AEs leading to discontinuation: 3 (NR) |
| Supportive care (N=85) | • SAEs: 1 (NR) | |
| TESTING/ NCT01560052 - Kim et al. [38][e] & Lv et al. [63][f] | | |
| Placebo (N=14) | • Any AE: 12 (85.7%)[e,f]<br>• Serious TEAE: 1 (7.1%)[f]<br>• Treatment discontinuation: 0 (0%)[f] | • TEAEs leading to death: 0 (0%)[f]<br>• AE resulting in dose reduction: 1 (7.1%)[f] |
| Telitacicept [160 mg] (N=16) | • Any AE: 15 (93.8%)[e,f]<br>• Serious TEAE: 1 (6.3%)[f]<br>• Treatment discontinuation: 0 (0%)[f] | • TEAEs leading to death: 0 (0%)[f]<br>• AE resulting in dose reduction: 1 (6.3%)[f] |
| Telitacicept [240 mg] (N=14) | • Any AE: 13 (92.9%)[e,f]<br>• Serious TEAE: 2 (14.3%)[f]<br>• Treatment discontinuation: 0 (0%)[f] | • TEAEs leading to death: 0 (0%)[f]<br>• AE resulting in dose reduction: 3 (21.4%)[f] |
| NCT02981212 - Han et al. [52]; | | |
| MMF + corticosteroid (N=24) | • Any AE: 23 (88.5%)<br>• SAEs: 2 (7.7%) | • TRAEs: 12 (46.2%) |
| Control (N=20) | • Any AE: 15 (68.2%)<br>• SAEs: 2 (9.1%) | • TRAE: 0 (0%) |
| ChiCTR1800014442 – Li et al. [90] | | |
| MP + alternative low-dose prednisone (N=45) | • Any AE: 28 (62%) | |
| Full-dose prednisone (N=42) | • Any AE: 30 (71%) | |
| NCT02160132 - Liang et al. [91] | | |
| MP + alternative low-dose prednisone (N=45) | • Any AE: 28 (62%) | |
| Full-dose prednisone (N=42) | • Any AE: 30 (71%) | |
| NCT03841448 - www.clinicaltrials.gov (93) | | |
| Cemdisiran (N=22) | • Any AE: 14 (70%)<br>• SAEs: 1 (5%) | |

*(Continued)*

**Table 6.** (Continued)

| Treatment (N) | AEs, N (%) | | |
|---|---|---|---|
| Placebo (N=9) | • Any AE: 8 (87.5%)<br>• SAEs: 1 (12.5%) | | |
| **ISRCTN97636235 – Ni et al. [53]** | | | |
| Leflunomide + prednisone (N=59) | • Any AE: NR (36%) | | |
| Prednisone (N=49) | • Any AE: NR (55%) | | |
| **NCT02942381 – Liu et al. [57]** | | | |
| Hydroxychloroquine sulfate (N=30) | • AEs leading to treatment discontinuation: 4 (NR) | • Any AE: 7 (NR)<br>• Serious AEs: 0 (0%) | |
| Placebo (N=30) | • AEs leading to treatment discontinuation: 0% | • Any AE: 2 (NR)<br>• Serious AEs: 0 (0%) | |
| **STOP-IgAN/ NCT00554502 – Rauen et al. [56] & Rauen et al. [12]** | | | |
| Supportive therapy (RAS blockade) (N=80) | • SAEs: 29 (NR)<br>• Total number of infection events: 111 (NR) | | |
| Supportive therapy (RAS blockade) + immunosuppression (N=82) | • SAEs: 33 (NR)<br>• Total number of infection events: 174 (NR) | | |
| High-eGFR: Supportive therapy (RAS blockade) (N=54) | • Total number of infection events: 69 (NR)<br>• Total SAEs of infection: 2 (NR)<br>• SAEs: 19 (NR) | | |
| High-eGFR: supportive therapy (RAS blockade) + immunosuppression (N=55) | • Total number of infection events: 115 (NR)<br>• Total SAEs of infection: 4 (NR)<br>• SAEs: 14 (NR) | | |
| Low eGFR: Supportive therapy (RAS blockade) (N=26) | • Total number of infection events: 48 (NR)<br>• Total SAEs of infection: 1 (NR)<br>• SAEs: 9 (NR) | | |
| Low eGFR: supportive therapy (RAS blockade) + immunosuppression (N=27) | • Total number of infection events: 59 (NR)<br>• Total SAEs of infection: 4 (NR)<br>• SAEs: 19 (NR) | | |
| **NEFIGAN/ NCT01738035 – Fellström et al. [58]** | | | |
| Placebo (N=50) | • TRAE/TEAEs leading to discontinuation: 2 (NR) | • Any TRAE/TEAEs: 42 (84%)<br>• Infection TRAEs/TEAEs: 3 (6%) | |
| TRF-B [8 mg/day] (N=51) | • TRAE/TEAEs leading to discontinuation: 5 (NR) | • Any TRAE/TEAEs: 48 (94%)<br>• Infection TRAEs/TEAEs: 2 (4%) | |
| TRF-B [16 mg/day] (N=48) | • TRAE/TEAEs leading to discontinuation: 11 (NR) | • Any TRAE/TEAEs: 43 (88%)<br>• Infection TRAEs/TEAEs: 6 (12%) | |
| **Hirai et al. [59]** | | | |
| Standard therapy (steroid pulse and tonsillectomy) + mizoribine (N=21) | • AEs leading to discontinuation: 2 (NR) | | |
| Standard therapy (steroid pulse and tonsillectomy) (N=21) | • NR | | |
| **NCT01269021 – Hou et al. [60]** | | | |
| MMF + prednisone (N=86) | • AEs leading to discontinuation: 2 (2%)<br>• Any AE 68 (78%) | • SAEs: 5 (6%)<br>• Any TRAE/TEAE: 54 (62%)<br>• Any AE: Infection: 27 (31%) | |
| Prednisone (N=88) | • AEs leading to discontinuation: 2 (2%)<br>• Any AE 68 (77%) | • SAEs: 6 (7%)<br>• Any TRAE/TEAE: 60 (68%)<br>• Any AE: Infection: 20 (23%) | |
| **Min et al. [64]** | | | |
| Prednisone (N=45) | • Any AE: 12 (30%)<br>• Any AE: Pulmonary infection: 2 (NR) | • Any AE: Upper respiratory tract infection: 4 (NR) | |
| Leflunomide + prednisone (N=40) | • Any AE: 13 (28.9%)<br>• Any AE: Pulmonary infection: 1 (NR) | • Any AE: Upper respiratory tract infection: 4 (NR) | |

*(Continued)*

**Table 6.** (Continued)

| Treatment (N) | AEs, N (%) | |
|---|---|---|
| **NCT01224028 – Kim et al. [66]** | | |
| Placebo (N=20) | • Treatment discontinuation (any reason): 1 (NR)<br>• AEs leading to discontinuation: 0 (0%)<br>• Any AE: 15 (NR) | • Mild AEs: 15 (NR)<br>• Moderate AEs: 0 (0%)<br>• Severe AEs: 0 (0%)<br>• Any TRAE/TEAE: 0 (0%) |
| Tacrolimus (N=20) | • Treatment discontinuation (any reason): 2 (NR)<br>• AEs leading to discontinuation: 1 (NR)<br>• Any AE: 49 (NR) | • Mild AEs: 43 (NR)<br>• Moderate AEs: 6 (NR)<br>• Severe AEs: 0 (0%)<br>• Any TRAE/TEAE: 16 (NR) |
| **UMIN000000593 – Masutani et al. [67]** | | |
| MP + prednisolone (N=20) | • Treatment discontinuation (any reason): 5 (NR) | |
| MP + prednisolone + mizoribine (N=20) | | |
| **NCT00318474 – Hogg et al. [68]** | | |
| MMF (N=7) | • AEs leading to discontinuation: 1 (NR) | |
| Placebo (N=10) | • SAEs: 1 (NR) | |
| **Liu et al. [69]** | | |
| MP + CSA (N=23) | • Any AE: 2 (4.76%) | |
| MP (N=25) | • Any AE: 11 (26.2%) | |
| **NCT00755859/ NCT01392833 – Pozzi et al. [120][g] & Pozzi et al. [121][h]** | | |
| MP + prednisone + azathioprine (N=101[a]/N=20[b]) | • Treatment discontinuation: 15 (NR)[g]/ 7 (35%)[h]<br>• Patients with at least 1 event: 43 (NR)[g]/ 16 (80%)[bh] | • Total number of events: 60[g]/ 21[h]<br>• AEs leading to discontinuation: 6 (30%)[h]<br>• Any TRAE/ TEAE: 17 (NR)[g]/ 6 (30%)[h]<br>• Any TRAE/TEAE: Bacteria infection: 3 (NR)[g,h] |
| MP + prednisone (N=106[a]/N=26[b]) | • Treatment discontinuation: 3 (NR)[g]/ 4 (15%)[h]<br>• Patients with at least 1 event: 38 (NR)[g]/ 14 (54%)[h] | • Total number of events: 44[g]/ 19[h]<br>• AEs leading to discontinuation: 4 (15%)[h]<br>• Any TRAE/ TEAE: 6 (NR)[g]/ 4 (15%)[h]<br>• Any TRAE/TEAE: Bacteria infection: 3 (NR)[g,h] |
| **NCT00863252 – Tang et al. [72][i] & Tang et al. [73][j]** | | |
| MMF (N=20) | • AEs leading to discontinuation: 0 (0%)[i] | • AEs leading to hospitalization: 0 (0%)[i] |
| Conventional therapy (N=20) | • NR | |
| **Lou et al. [77]** | | |
| Leflunomide (N=24) | • AEs leading to discontinuation: 1 (NR)<br>• Any AE: 4 (16%) | • SAEs: 1 (4%) |
| Fosinopril (N=22) | • AEs leading to discontinuation: 0 (0%)<br>• Any AE: 2 (9%) | • SAEs: 0 (0%) |
| **Frisch et al. [78]** | | |
| MMF (N=17) | • TRAEs/ TEAEs leading to death: 0 (0%)<br>• Treatment discontinuation (any reason): 2 (NR) | • TRAEs/ TEAEs leading to discontinuation: 2 (NR) |
| Placebo (N=15) | • TRAEs/ TEAEs leading to death: 0 (0%)<br>• Treatment discontinuation (any reason): 2 (NR) | • TRAEs/ TEAEs leading to discontinuation: 2 (NR) |
| **ENVISION/ NCT04287985 – Mathur et al. [97]** | | |
| Sibeprenlimab [2 mg/kg] (N=38) | • Any AE: 28 (73.7%)<br>• Mild AE: 19 (50%)<br>• Moderate AE: 7 (18.4%)<br>• Severe AE: 2 (5.3%)<br>• SAE: 2 (5.3%)<br>• Infection AEs: 15 (39.5%)<br>• TRAEs: 7 (18.4%) | • AEs leading to death: 0 (0%)<br>• Treatment discontinuation (any reason): 1 (2.6%)<br>• AE resulting in dose reduction: 5 (13.2%) |

*(Continued)*

**Table 6.** (Continued)

| Treatment (N) | AEs, N (%) | |
|---|---|---|
| Sibeprenlimab [4 mg/kg] (N=41) | • Any AE: 33 (80.5%)<br>• Mild AE: 22 (53.7%)<br>• Moderate AE: 9 (22.0%)<br>• Severe AE: 2 (4.9%)<br>• SAE: 4 (4.9%)<br>• Infection AEs: 23 (56.1%)<br>• TRAEs: 7 (17.1%) | • AEs leading to death: 0 (0%)<br>• Treatment discontinuation (any reason): 0 (0%)<br>• AE resulting in dose reduction: 1 (2.4%) |
| Sibeprenlimab [8 mg/kg] (N=38) | • Any AE: 31 (81.6%)<br>• Mild AE: 22 (57.9%)<br>• Moderate AE: 8 (21.1%)<br>• Severe AE: 1 (2.6%)<br>• SAE: 1 (2.6%)<br>• Infection AEs: 20 (52.6%)<br>• TRAEs: 4 (10.5%) | • AEs leading to death: 0 (0%)<br>• Treatment discontinuation (any reason): 0 (0%)<br>• AE resulting in dose reduction: 3 (7.9%) |
| Placebo (N=38) | • Any AE: 27 (71.1%)<br>• Mild AE: 23 (60.5%)<br>• Moderate AE: 3 (7.9%)<br>• Severe AE: 1 (2.6%)<br>• SAE: 2 (5.3%)<br>• Infection AEs: 21 (55.3%)<br>• TRAEs: 5 (13.2%) | • AEs leading to death: 1 (2.6%)<br>• Treatment discontinuation (any reason): 0 (0%)<br>• AE resulting in dose reduction: 0 (0%) |
| NCT02112838 - Tam et al. [96] | | |
| Placebo (N=25) | • Any TEAE: 21 (84%)<br>Mild TEAEs: 15 (60%)<br>Moderate TEAEs: 5 (20%)<br>Severe TEAEs: 1 (4%)<br>• Serious TEAEs: 2 (8%) | • TRAEs: 0 (0%)<br>TEAEs leading to death: 0 (0%)<br>• Deaths: 0 (0%)<br>• TEAEs leading to discontinuation: 1 (4%) |
| Fostamatinib [100 mg] (N=26) | • Any TEAE: 22 (85%)<br>Mild TEAEs: 14 (54%)<br>Moderate TEAEs: 8 (31%)<br>Severe TEAEs: 0 (0%)<br>• Serious TEAEs: 2 (8%) | • TRAEs: 1 (4%)<br>TEAEs leading to death: 0 (0%)<br>• Deaths: 0 (0%)<br>• TEAEs leading to discontinuation: 3 (12%) |
| Fostamatinib [150 mg] (N=25) | • Any TEAE: 24 (96%)<br>Mild TEAEs: 16 (64%)<br>Moderate TEAEs: 5 (20%)<br>Severe TEAEs: 3 (12%)<br>• Serious TEAEs: 2 (8%) | • TRAEs: 1 (4%)<br>TEAEs leading to death: 1 (4%)<br>• Deaths: 1 (4%)<br>• TEAEs leading to discontinuation: 7 (28%) |
| Manno et al. [75] | | |
| Ramipril (N=49) | • SAEs: 0 (0%) | |
| Prednisone + ramipril (N=48) | • SAEs: 0 (0%) | |
| Katafuchi et al. [82] | | |
| Prednisolone (N=43) | • AEs leading to discontinuation: 1 (NR) | |
| Control (N=47) | • NR | |
| Non-immunosuppressive therapies | | |
| PROTECT/ NCT03762850 – Heerspink et al. [22]k & Rovin et al. [18]l | | |
| Sparsentan (N=202) | • All TEAEs: 187 (93%)l<br>• SAEs: 43 (21%)k<br>• Serious TEAE: 75 (37%)l | • Deaths: 0 (0%)l<br>• TEAEs leading to death: 0 (0%)l<br>• TEAE leading to discontinuation: 21 (10%)l |
| Irbesartan (N=202) | • All TEAEs: 177 (88%)l<br>• SAEs: 41 (20%)k<br>• Serious TEAE: 71 (35%)l | • Deaths: 1 (<1%)l<br>• TEAEs leading to death: 1 (<1%)l<br>• TEAE leading to discontinuation: 18 (9%)l |

*(Continued)*

**Table 6.** (Continued)

| Treatment (N) | AEs, N (%) | |
|---|---|---|
| **DAPA-CKD/ NCT03036150 – Wheeler et al. [98]** | | |
| Dapagliflozin (N=137) | • SAEs: 22 (16.1%)<br>• Treatment discontinuation (any reason): 6 (4.4%)<br>AEs leading to discontinuation: 6 (4.4%) | |
| Placebo (N=133) | • SAEs: 34 (25.6%)<br>• Treatment discontinuation (any reason): 7 (5.3%)<br>AEs leading to discontinuation: 7 (5.3%) | |
| **NCT00793585 – Shi et al. [100]** | | |
| Allopurinol group (N=21) | • Treatment discontinuation (any reason): 3 (NR)<br>• Dose reduction: 4 (NR)<br>• AEs leading to discontinuation: 1 (NR) | |
| Control group (N=19) | • Treatment discontinuation (any reason): 2 (NR)<br>• AEs leading to discontinuation: 1 (NR) | |
| **Combination therapies** | | |
| **ChiCTR-TRC-10000776 – Wu et al. [105]** | | |
| Telmisartan (N=100) | • AEs leading to discontinuation: 1 (NR)<br>• SAEs: 0 (0%) | • Any AE: 4 (NR) |
| Telmisartan + clopidogrel (N=100) | • AEs leading to discontinuation: 1 (NR)<br>• SAEs: 0 (0%) | • Any AE: 7 (NR) |
| Telmisartan + leflunomide (N=100) | • AEs leading to discontinuation: 2 (NR)<br>• SAEs: 0 (0%) | • Any AE: 4 (NR) |
| Telmisartan + clopidogrel + leflunomide (N=99) | • AEs leading to discontinuation: 3 (NR)<br>• SAEs: 0 (0%) | • Any AE: 9 (NR) |
| **CRG030600070 – Xie et al. [107]** | | |
| Mizoribine + losartan (N=34) | • Any AE: 8 (23.53%) | |
| Mizoribine (N=35) | • Any AE: 6 (17.14%) | |
| Losartan (N=30) | • Any AE: 7 (23.33%) | |
| **Yoshikawa et al. [110]** | | |
| Prednisolone + azathioprine + heparin-warfarin + dipyridamole (N=40) | • Treatment discontinuation (any reason): 2 (NR) | |
| Heparin-warfarin and dipyridamole (N=38) | • Treatment discontinuation (any reason): 1 (NR) | |

**Abbreviations**: AE, adverse event; eGFR, glomerular filtration rate; MMF, mycophenolate mofetil; MP, methylprednisone; SAE, serious adverse event; TEAE, treatment emergent adverse events; TFR, targeted-release formulation; TRAE, treatment-related adverse events.

a,b In the NCT03373461 trial denotes which publication outcomes were reported; c,d In the NefIgArd/ NCT03643965 trial denotes which publication outcomes were reported; e,f In the TESTING/ NCT01560052 trial denotes which publication outcomes were reported; g,h In the NCT00755859/ NCT01392833 trial denotes which publication outcomes were reported; i,j In the NCT00863252 trial denotes from which publication the outcomes were reported; k,l In PROTECT/ NCT03762850 trial denotes which publication outcomes were reported.

and 28 reported overall AE rates (Table 6). Thirteen studies measured the effect of immunosuppressive/immunomodulatory therapies on PCR, 5 reported a significant reduction in PCR from baseline to follow-up [14,67,82,83,97]; 2 reported significantly lower PCR in the treatment group than the control/placebo group at follow-up [29,58] and 3 studies reported no significant differences between treatment and control groups at follow-up (Table 7) [12,68,95,96]. Kim et al. [66] and Yu et al. [65] reported the initial treatment and long-term follow-up phase, respectively, of a study in which patients treated with tacrolimus had a significantly lower PCR than the control group at the end of the 4-month treatment phase [66]. However, this effect was not maintained over 5-year follow-up, and PCR at final follow-up was not significantly lower than at baseline. Tam et al. [93,94] reported that after 6 months of treatment with fostamatinib, there was no significant difference

**Table 7. Proteinuria outcomes for patients treated with immunosuppressive/immunomodulatory therapy.**

| Author | Treatment (N) | FU, mo[a] | Age, yrs[b] | Baseline SCr, mg/dL[b] | Baseline Proteinuria[bc] | FU Proteinuria[bc] | Change from baseline[b] | p-value |
|---|---|---|---|---|---|---|---|---|
| Mathur et al. [97]; ENVISION/ NCT04287985 | Sibeprenlimab [2mg/kg] (N=38) | 16 | 41 (25, 71)[f] | NR | 1.46 (0.12) g/g, 1.47 (0.67, 6.92) g/day[d] | NR | -36.5% (10.6) [PCR], -0.68 (0.2) g/day | • Significant change from baseline at 12 months in treatment groups (primary endpoint) |
| | Sibeprenlimab [4mg/kg] (N=41) | 16 | 39 (20, 73)[f] | NR | 1.53 (0.12) g/g, 1.93 (0.33, 8.60) g/day[d] | NR | -58% (6.6) [PCR], -0.86 (0.2) g/day | |
| | Sibeprenlimab [8mg/kg] (N=38) | 16 | 42 (23, 72)[f] | NR | 1.44 (0.14) g/g, 1.90 (0.76, 12.44) g/day[d] | NR | -64.6% (5.7) [PCR], -1.06 (0.2) g/day | |
| | Placebo (N=38) | 16 | 36 (18, 52)[f] | NR | 1.68 (0.17) g/g, 2.13 (0.76, 8.48) g/day[d] | NR | -10.6% (15.0) [PCR], -0.21 (0.2) g/day | |
| Trial record, www.clinicaltrials.gov (NCT03373461)[87] & Zhang et al. [135]; NCT03373461 | LNP023 [10mg] (N=20) | 6 | 39.2 (12.42) | NR | 1.9 (1.1) g/g, 66.0 (28.5) g/day | NR | 1.06 (0.803, 1.394) [PCR][e], -16% (NR) [PCR][f], 0.80 (0.66, 0.97) [24h-PER][e], Compared to placebo: 0.95 (0.74, 1.22) [24h-PER][e] | • NR |
| | LNP023 [50mg] (N=19) | 6 | 36.6 (8.42) | NR | 1.7 (0.8) g/g, 53.8 (22.7) g/day | NR | 0.59 (0.452, 0.779) [PCR][e], -29% (NR) [PCR][df], 0.89 (0.74, 1.07) [24h-PER][e], Compared to placebo: 1.06 (0.83, 1.36) [24h-PER][e] | |
| | LNP023 [100mg] (N=22) | 6 | 36.0 (13.15) | NR | 1.8 (0.9) g/g, 67.0 (31.8) g/day | NR | 0.66 (0.540, 0.798) [PCR][e], -35% (NR) [PCR][f], 0.61 (0.51, 0.73) [24h-PER][e], Compared to placebo: 0.73 (0.57, 0.93) [24h-PER][e] | |
| | LNP023 [200mg] (N=26) | 6 | 42.5 (15.76) | NR | 1.3 (1.0) g/g, 57.9 (28.9) g/day | NR | 0.73 (0.568, 0.940) [PCR][e], -40% (16, 53) [PCR][f], 0.70 (0.60, 0.82) [24h-PER][e], Compared to placebo: 0.84 (0.67, 1.05) [24h-PER][e] | |
| | Placebo (N=25) | 6 | 39.4 (11.0) | NR | 1.3 (0.6) g/g, 65.7 (32.6) g/day | NR | 0.91 (0.705, 1.185) [PCR][e], -2% (NR)[f], 0.84 (0.71, 0.99) [24h-PER][e] | |
| Barratt et al. [34][g]; NCT03373461 | Iptacopan [200mg] (N=26) | 6 | NR | NR | NR | NR | -41% (31, 49) [PCR][e] | • NR |
| | Placebo (N=25) | 6 | NR | NR | NR | NR | -2% (-20, 23) [PCR][e] | |

*(Continued)*

| Author | Treatment (N) | FU, mo[a] | Age, yrs[b] | Baseline SCr, mg/dL[b] | Baseline Proteinuria[bc] | FU Proteinuria[bc] | Change from baseline[b] | p-value |
|---|---|---|---|---|---|---|---|---|
| Barratt et al. [16]; NeflgArd/ NCT03643965 | TRF-B (N=97) | 9 | 44 (25-69)[d] | 1.27 (0.95-1.75)[h] | NR | NR | -31% (NR) [PCR][i] vs placebo: 27% (-39, -13) [PCR][i] | • p<0.0003 change from baseline vs placebo |
| | Placebo (N=102) | 9 | 43 (23-73)[d] | 1.21 (0.87-1.79)[h] | NR | NR | -5% (NR) [PCR][i] | • NR |
| Lafayette et al. [17]; NeflgArd/ NCT03643965 | TRF-B [16mg/day] (N=182) | 24 | 43 (36-50)[d] | 1.28 (0.9-1.76)[h] | NR | NR | 24 months: -30.7% (-38.9, -21.5) [PCR][i] | • p<0.0001 between arms |
| | Placebo (N=182) | 24 | 42 (34-49)[d] | 1.25 (0.88-1.74)[h] | NR | NR | -1.0% (-12.8, 12.4) [PCR][i] | |
| Barratt et al. [33]; NCT04564339 | Ravulizumab (N=43) | 6 | 40.1 (NR) | NR | NR | NR | -40.3% (NR) [24h-PER] | • p=0.0012 between arms |
| | Placebo (N=23) | 6 | | NR | NR | NR | -10.9% (NR) [24h-PER] | |
| Lafayette et al. [35]; ORIGIN/ NCT04716231 | Atacicept [75mg & 150 mg] (N=66) | 6 | NR | NR | NR | NR | -31% (NR) [PCR] | • p=0.037 vs placebo |
| | Atacicept [150 mg] (N=33) | 6 | NR | NR | NR | NR | -33% (NR) [PCR] | • p=0.047 vs placebo |
| | Atacicept [75 mg] (N=33) | 6 | NR | NR | NR | NR | NR | • NS vs placebo |
| | Atacicept [25 mg] (N=16) | 6 | NR | NR | NR | NR | NR | • NS vs placebo |
| | Placebo (N=34) | 6 | NR | NR | NR | NR | -7% (NR)[PCR] | Ref |
| Barratt et al. [36]; ORI-GIN/ NCT04716231 | Atacicept [150 mg] (N=33) | 9 | NR | NR | NR | NR | -33% (NR)[PCR] | • p=0.012 vs placebo |
| | Placebo (N=34) | 9 | NR | NR | NR | NR | 3% (NR)[PCR] | |
| Hou et al. [89]; MAIN/ NCT0185481 | MMF (N=85) | 36 | 35 (8.7) | NR | 2.1 (1.9) g/day | 1.2 (NR) g/day | -57.1% (-85.1, 0) [24h-PER][i] | • p<0.001 between treatment arms at FU |
| | Supportive care (N=85) | 36 | 38.2 (9.8) | NR | 1.7 (1.3) g/day | **0.5 (NR) g/day** | -28.2% (52.2, 60.7) [24h-PER][i] | • p<0.001 change in PU24 between treatment arms |
| Kim et al. [38]; TEST-ING/ NCT01560052 | MP (N=121) | 12 | 37 (NR) | NR | 2.48 (NR) g/day | 1.58 (NR) g/day | Difference between arms: 1.15 g/day (NR) | • p=0.0002 between arms at FU |
| | Placebo (N=120) | 12 | 37 (NR) | NR | | 2.41 (NR) g/day | | |

*(Continued)*

| Author | Treatment (N) | FU, mo[a] | Age, yrs[b] | Baseline SCr, mg/dL[b] | Baseline Proteinuria[bc] | FU Proteinuria[bc] | Change from baseline[b] | p-value |
|---|---|---|---|---|---|---|---|---|
| Lv et al. [63]; TESTING/ NCT01560052 | MP (N=257) | 3.5 years[h] | 35.6 (29.4, 46.3)[h] | NR | 1.99 (13.6, 3.09) g/day[i] | 1.70 (1.54, 1.86) g/day[i] | NR | • p<0.001 between arms at FU |
| | Placebo (N=246) | 3.5 years[h] | 36.6 (29, 45.9)[h] | NR | 1.93 (1.38, 2.88) g/day[i] | 2.39 (2.15, 2.63) g/day[i] | NR | |
| | Full-dose protocol MP (N=136) | 3.5 years[h] | 36.5 (29, 46.5)[h] | NR | 2.108 (1.475, 3.033) g/day[i] | 1.8 (1.57, 2.03) g/day[i] | NR | • P=0.003 between arms at FU |
| | Full-dose protocol placebo (N=126) | 3.5 years[h] | 37 (28, 47)[h] | NR | 1.928 (1.49, 2.865) g/day[i] | 2.38 (2.07, 2.68) g/day[i] | NR | |
| | Reduced dose protocol MP (N=121) | 3.5 years[h] | 35 (28, 44)[h] | NR | 1.975 (1.39, 2.960) g/day[i] | 1.58 (1.36, 1.8) g/day[i] | NR | • p<0.001 between arms at FU |
| | Reduced dose protocol placebo (N=120) | 3.5 years[h] | 36 (28.5, 44.5)[h] | NR | 2.012 (1.496, 3.055) g/day[i] | 2.41 (2.04, 2.78) g/day[i] | NR | |
| Lv et al. [54]; NCT04291781 | Placebo (N=14) | 6 | 38.3 (6.9) | NR | 1.95 (0.21) g/day | 1.9 (0.30) g/day | NR | • NR |
| | Telitacicept 160mg (N=16) | 6 | 35.9 (9.9) | NR | 1.97 (0.27) g/day | 1.66 (0.41) g/day | -0.29 g/day (-0.95, 0.37)[i] -25% (NR) [24h-PER] | • p=0.389 FU vs baseline |
| | Telitacicept 240mg (N=14) | 6 | 36.8 (8.8) | NR | 1.65 (0.22) g/day | 0.76 (0.14) g/day | -0.889 g/day (-1.57, -0.2)[i] -49% (NR) [24h-PER] | • \p=0.013 FU vs baseline |
| Sun et al. [92] | Supportive care (N=71) | 9 | 39.3 (10.1) | 120.2 (30.6) µmol/L | 1.9 (0.4) g/day | 1.8 (0.9, 2.6) g/day[h] | NR | • p=0.002 between treatment arms at FU |
| | Supportive care + fluticasone (N=71) | 9 | 39.6 (10.5) | 127.5 (35.7 µmol/L | 1.8 (0.7) g/day | 0.9 (0.5, 1.0) g/day[h] | NR | |

(Continued)

**Table 7.** (Continued)

| Author | Treatment (N) | FU, mo[a] | Age, yrs[b] | Baseline SCr, mg/dL[b] | Baseline Proteinuria[bc] | FU Protein-uria[bc] | Change from baseline[b] | p-value |
|---|---|---|---|---|---|---|---|---|
| Tam et al. [96] & Trial record for NCT02112838 | Placebo (N=14) | 6 | 40.5 (NR)[d] | NR | 1.272 (0.525, 9.38) g/g[d] | 1.034 (0.078, 13.819) g/g[d] | -0.177 (NR) g/g[d] | • NR |
| | Fostamatinib [100 mg] (N=16) | 6 | | NR | 1.828 (0.387, 16.259) g/g[d] | 0.842 (0.097, 9.803) g/g[d] | -0.720 (NR) g/g[d] | • NR |
| | Fostamatinib [150 mg] (N=15) | 6 | | NR | 1.878 (0.664, 4.076) g/g[d] | 1.299 (0.309, 4.661) g/g[d] | -0.803 (NR) g/g[d] | • NR |
| | Fostamatinib [150 mg] (N=25) | 6 | 43.1 (14.8) | NR | NR | NR | -0.157 (0.345) g/g -8% (NR) [PCR] | • p=0.97 150mg vs placebo at FU |
| | Fostamatinib [100 mg] (N=26) | 6 | 42.3 (14.1) | NR | NR | NR | -0.577 (0.335) g/g -25% (NR) [PCR] | • p=0.4 100mg vs placebo at FU |
| | Placebo (N=25) | 6 | 40.6 (11.6) | NR | NR | NR | -0.177 (0.342) g/g -9% (NR) [PCR] | |
| Trial record, [93]; NCT03841448 | Cemdisiran (N=22) | 8 | 40.5 (10.1) | NR | NR | NR | 0.686 (0.098) [PCR] 0.671 (0.104) [24h-PER] | • NR |
| | Placebo (N=9) | 8 | 37.6 (10.4) | NR | NR | NR | 1.095 (0.258) [PCR] 1.051 (0.266) [24h-PER] | |
| Han et al. [52]; NCT02981212 | MMF + corticosteroid (N=24) | 12 | 44.0 (10.6) | 1.71 (0.56) | 1.71 (0.56) g/g | 1.27 (0.52) g/g | -0.47 (0.17) g/g | • P=0.01 vs control at FU [PCR] • p=0.04 averaged change in PCR vs control |
| | Control/ supportive care (N=20) | 12 | 46.1 (7.8) | 2.26 (0.91) | 2.26 (0.91) g/g | 1.97 (0.89) g/g | 0.07 (0.17) g/g | |
| Li et al. [90]; ChiCTR1800014442 | MP + alternative low-dose prednisone (N=45) | 18 | 35 (31, 39)[h] | 65.0 (54.5, 78.3) µmol/L[h] | 2 (0.75) g/ day | NR | -1.55 (-1.77, -1.33) g/day[h] | • p=0.604 change in PU24 between treatment arms |
| | Full-dose prednisone (N=42) | 18 | 36 (31, 41)[h] | 73.7 (58.5, 84.2) µmol/L[h] | 1.99 (0.77) g/day | NR | -1.55 (-1.83, -1.28) g/day[h] | |
| Liang et al. [136]; NCT02160132 | 1–2-3 MP group (N=34) | 6 | 29.7 (10.5) | 98.4 (38.2) µmol/L | 2.04 (1.81) g/day | **0.64 (1.09) g/ day** | NR | • p<0.001 for all treatments at FU vs baseline |
| | 1–3-5 MP group (N=34) | 6 | 33.8 (9.9) | 105.5 (47.2) µmol/L | 1.74 (0.93) g/day | **0.73 (0.79) g/ day** | NR | |
| | All (N=68) | 6 | 31.7 (10.3) | 102.0 (42.7) µmol/L | 1.89 (1.43) g/day | **0.68 (0.95) g/ day** | NR | |

*(Continued)*

Table 7. (Continued)

| Author | Treatment (N) | FU, mo[a] | Age, yrs[b] | Baseline SCr, mg/dL[b] | Baseline Proteinuria[bc] | FU Proteinuria[bc] | Change from baseline[b] | p-value |
|---|---|---|---|---|---|---|---|---|
| Zhang et al. [39][g] | Leflunomide + low-dose prednisone (N=59) | 12 | NR | NR | NR | NR | NR | • p<0.01, decrease in 24PU from baseline to FU in both treatment arms • p>0.05 difference at follow-up between treatment arms |
| | Prednisone alone (N=49) | 12 | NR | NR | NR | NR | NR | |
| Ni et al. [53]; ISRCTN97636235 | Leflunomide [20-40 mg/day] + prednisone [0.5-0.8 mg/kg/day] (N=59) | 24 | 35.7 (11.2) | 99.3 (56.8) μmol/L | 1.8 (1.3, 3.5) g/day[h] | **0.5 (0.1, 1.1) g/ day[h]** | NR | • p<0.01 from baseline to FU (leflunomide + prednisone) |
| | Prednisone [0.5-0.8 mg/kg/day] (N=49) | 24 | 35.5 (11.2) | 96.4 (38.6) μmol/L | 1.9 (1.2, 2.9) g/day[h] | **0.5 (0.2, 1.1) g/ day[h]** | NR | • p<0.01 from baseline to FU (prednisone) |
| Lennartz et al. [55] & Rauen et al. [12]; STOP-IgAN/ NCT00554502 | Single RAS blockade (N=41) | 36 | 45.5 (12.2) | NR | 1.0 (0.5) g/g | 0.8 (0.6) g/g | NR | • NR |
| | Single RAS blockade + immunosuppression (N=37) | 36 | | | 1.1 (0.6) g/g | 0.6 (0.6) g/g | NR | |
| | Dual RAS blockade (N=16) | 36 | 44.5 (12.8) | NR | 0.9 (0.5) g/g | 1.1 (0.8) g/g | NR | |
| | Dual RAS blockade + immunosuppression (N=14) | 36 | | | 1.2 (0.7) g/g | 1.0 (1.2) g/g | NR | |
| | High eGFR[k]: supportive therapy (RAS blockade) (N=54) | 36 | 45.6 (11.9) | 1.4 (0.5) | 0.9 (0.5) g/g | 0.80 (0.64) g/g | NR | • p=0.26 supportive therapy vs supportive therapy + immunosuppression |
| | High eGFR[k]: supportive therapy (RAS blockade) + immunosuppression (N=55) | 36 | 41.7 (13.3) | 1.3 (0.4) | 0.9 (0.5) g/g | 0.57 (0.53) g/g | NR | |
| | Low eGFR[k]: Supportive therapy (RAS blockade) (N=26) | 36 | 46.0 (14.0) | 2.0 (0.6) | 1.1 (0.6) g/g | 0.98 (0.71) g/g | NR | • p=0.35 supportive therapy vs supportive therapy + immunosuppression |
| | Low eGFR[k]: supportive therapy (RAS blockade) + immunosuppression (N=27) | 36 | 45.1 (12.8) | 2.2 (0.7) | 1.5 (0.7) g/g | 1.27 (1.4) g/g | NR | |
| Liu et al. [57]; NCT02942381 | Hydroxychloroquine sulfate [400 mg/day] (N=30) | 6 | 37.6 (11.6) | 127.9 (41.9) μmol/L | 1.6 (1.1, 2.2) g/day[h] | **0.9 (0.6, 1.0) g/ day[h]** | -48.4% (-64.2, -30.5) [24h-PER][h] | • p=0.002 between treatment arms • p<0.001 change from baseline between study arms |
| | Placebo (N=30) | 6 | 35.6 (9.6) | 120.2 (32.8) μmol/L | 1.9 (1.3, 2.6) g/day[h] | 1.9 (0.9, 2.6) g/ day[h] | 10% (-38.7, 30.6) [24h-PER][h] | |

*(Continued)*

| Author | Treatment (N) | FU, mo[a] | Age, yrs[b] | Baseline SCr, mg/dL[b] | Baseline Proteinuria[bc] | FU Proteinuria[bc] | Change from baseline[b] | p-value |
|---|---|---|---|---|---|---|---|---|
| Fellström et al. [58]; NEFIGAN/ NCT01738035 | Placebo (N=50) | 12 | 38.9 (12.0) | NR | 0.8 (0.5, 1.6) g/g[h] | NR | 0.5%[j] | • p=0.0101 change in PCR 8mg vs placebo. |
| | | | | | 1.2 (1.0, 3.2) g/day[h] | | NR | • p=0.0005 change in PCR, 16mg vs placebo. |
| | TRF-B [8 mg/day] (N=51) | 12 | 40.6 (13.0) | NR | 0.8 (0.5, 1.2) g/g[h] | NR | -22.6% [PCR][j] | • p=0.0085 from baseline to FU (24h-PER, 8mg) |
| | | | | | 1.1 (0.9, 1.8) g/day[h] | 0.764 (0.613, 0.952) g/day[h] | NR | • p<0.0001 from baseline to FU (24h-PER, 16mg) |
| | TRF-B [16 mg/day] (N=48) | 12 | 37.5 (11.9) | NR | 0.8 (0.5, 1.3) g/g[h] | NR | -32% [PCR][j] | |
| | | | | | 1.1 (0.9, 1.8) g/day[h] | 0.764 (0.613, 0.952) g/day[h] | NR | |
| Lafayette et al. [61]; NCT00498368 | Rituximab [1g] + standard therapy (fish oil with ACEi/ ARBs) (N=17) | 12 | 43 (29, 63)[d] | 1.7 (0.8, 2.3)[h] | 2.6 (0.9, 5.3) g/day[d] | 1.9 (0.4, 8.8) g/day[d] | NR | • p=0.30 baseline to FU (rituximab + standard therapy) |
| | Standard therapy (fish oil with ACEi/ARBs) (N=17) | 12 | 33 (21, 59)[d] | 1.3 (0.8, 2.4)[h] | 1.8 (0.5, 4.0) g/day[d] | 1.8 (0.4, 4.3) g/day[d] | NR | |
| Min et al. [64] | Prednisone [0.8 mg/kg/day] (N=45) | 60 | 3.6 (11.5) | 95.1 (31.6) µmol/L | 2.16 (1.36, 3.5) g/day[h] | 0.96 (0.32, 1.75) g/day[h] | NR | • P<0.05 between treatment arms |
| | Leflunomide [20-40 mg/day] + prednisone [0.8 mg/kg/day] (N=40) | 60 | 36.9 (10.5) | 92.4 (34.0) µmol/L | 1.94 (1.21, 2.87) g/day[h] | 0.27 (0.15, 1.03) g/day[h] | NR | |
| Yu et al. [65] & Kim et al. [66]; NCT01224028 | Tacrolimus [0.1 mg/kg/day] (N=20) [FU phase] | 57.9 (13.8) | 36.9 (11.4) | NR | 0.649 (0.452) g/g, after treatment phase | 1.611 (1.410) g/g | NR | • p=NS baseline of trial phase vs final FU (Tacrolimus) |
| | Placebo (N=20) [FU phase] | 57.9 (13.8) | 41.0 (2.6) | NR | 0.998 (0.479) g/g, after treatment phase | 1.33 (0.726) g/g | NR | |
| | Placebo (N=20) [treatment phase] | 4 | 40.1 (12.8) | 0.98 (0.26) | 1.202 (0.5) g/g | 0.973 (0.471) g/g | -49.7% (21.9%) [PCR] | • p=0.033 between study arms at FU |
| | Tacrolimus [0.1 mg/kg/day] (N=20) [treatment phase] | 4 | 36.9 (11.4) | 1.06 (0.30) | 1.398 (0.809) g/g | 0.863 (0.798) g/g | -14.4% (40.0%) [PCR] | |

*(Continued)*

| Author | Treatment (N) | FU, mo[a] | Age, yrs[b] | Baseline SCr, mg/dL[b] | Baseline Proteinuria[bc] | FU Protein-uria[bc] | Change from baseline[b] | p-value |
|---|---|---|---|---|---|---|---|---|
| Masutani et al. [67]; UMIN000000593 | MP [500 mg] + prednisolone [30 mg/day] (N=20) | 25 | 36.4 (12.9) | 0.9 (0.38) | 0.98 (0.56, 1.91) g/g[h] | 0.17 (0.07, 0.61) g/g[h] | NR | • p<0.01 from base-line to FU (MP + prednisolone) • p<0.01 from baseline to FU (MP + prednisolone + mizoribine) |
| | MP [500 mg] + prednisolone [30 mg/day] + mizoribine [150 mg/day] (N=20) | 25 | 43.8 (10.8) | 0.96 (0.35) | 1.01 (0.50, 1.85) g/g[h] | 0.38 (0.19, 0.72) g/g[h] | NR | |
| Hogg et al. [68]; NCT00318474 | MMF [25-36 mg/kg/day] (N=7 at FU) | 24 | 31.8 (11.7) | NR | 1.25 (0.94, 1.55) g/g[h] | 1.22 (0.70, 1.74) g/g[h] | -0.03 (-0.58, 0.52) g/g[h] | • p=0.6 MMF vs placebo at FU |
| | Placebo (N=10 at FU) | 24 | 32.2 (13.2) | NR | 1.44 (1.00, 1.88) g/g[h] | 1.67 (0.53, 2.82) g/g[h] | 0.24 (-0.72, 1.19) g/g[h] | |
| Liu et al. [69] | MP [0.8 mg/kg/day] + CSA [3 mg/kg/day] (N=23) | 12 | 42.4 (13.1) | NR | 2.60 (2.03) g/day | **0.36 (0.23) g/day** | NR | • p<0.001 from baseline to FU (MP+CSA) • p<0.001 from base-line to FU (MP) |
| | MP [0.8 mg/kg/day] (N=25) | 12 | 36.8 (8.1) | NR | 3.17 (3.25) g/day | **0.53 (0.71) g/day** | NR | |
| Liu et al. [70] | MMF [0.75-1 g/day] + prednisone [0.8-1mg/kg/day] (N=42) | 18 | 29.8 (3.8) | 1.01 (0.26) | 2.83 (0.65) g/day | **0.6 (0.3) g/day** | NR | • p<0.01 from base-line to FU (MMF + prednisone) • p<0.01 from baseline to FU (cyclophosphamide + prednisone) • p<0.05 between treatment arms |
| | Cyclophosphamide [0.8-1 g/month] + prednisone [0.8-1mg/kg/day] (N=42) | 18 | 37.4 (4.8) | 1.02 (0.28) | 2.77 (0.81) g/day | 1.4 (0.5) g/day | NR | |
| Pozzi et al. [121] & Pozzi et al. [120]; NCT00755859/ NCT01392833 | MP [1 g] + prednisone [0.5 mg/kg] + azathioprine [1.5 mg/kg/day] (N=20) | 54 (34.8, 73.2)[h] | 43.0 (32.6, 52.4)[h] | 2.6 (2.37, 3.04)[h] | 3.20 (1.74, 5.54) g/day[h] | 2.73 (0.83, 4.13) g/day[h] | NR | • p<0.001 between study arms at FU |
| | MP [1 g] + prednisone [0.5 mg/kg] (N=26)[121] | 54 (34.8, 73.2)[h] | 37.3 (32.7, 52.3)[h] | 2.85 (2.38, 2.55)[h] | 2.00 (1.50, 3.23) g/day[h] | 1.05 (0.53, 1.47) g/day[h] | NR | |
| | MP [1 g] + prednisone [0.5 mg/kg] + azathioprine [1.5 mg/kg/day] (N=101) | 58.8 (36, 16.8)[h] | 34.8 (27.7, 43.9)[h] | 1.2 (1.0, 1.5)[h] | 2.1 (NR) g/day[h] | 1.16 (NR) g/day[h] | -44.8% (NR)[24-PER] | • p=0.57 between treatment arms at FU |
| | MP [1 g] + prednisone [0.5 mg/kg] (N=106) | 58.8 (36, 16.8)[h] | 40.5 (30.3, 51.3)[h] | 1.28 (1.0, 1.66)[h] | 1.95 (NR) g/day[h] | **0.98 (NR) g/day[h]** | -49.9% (NR)[24h-PER] | • p<0.01 change from baseline between treatment arms |

*(Continued)*

Table 7. (Continued)

| Author | Treatment (N) | FU, mo[a] | Age, yrs[b] | Baseline SCr, mg/dL[b] | Baseline Proteinuria[bc] | FU Proteinuria[bc] | Change from baseline[b] | p-value |
|---|---|---|---|---|---|---|---|---|
| Liu et al. [71] | Prednisone [0.8mg/kg/day] + leflunomide [20-50mg/day] (N=20) | 6 | 30.4 (16.2) | 96.4 (24.6) | 4.8 (2.6) g/day | 1.06 (0.28) g/day | NR | • p<0.01 from baseline to FU (prednisone + leflunomide) |
| | Prednisone [0.8mg/kg/day] + MMF [1g/day] (N=20) | 6 | 32.1 (14.6) | 92.8 (26.1) | 4.9 (2.4) g/day | 1.04 (0.31) g/day | NR | • p<0.01 from baseline to FU (predni-sone + MMF) |
| Tang et al. [73] & Tang et al. [72]; NCT00863252 | MMF [1.5-2g/day] (N=20) | 18 | 42.1 (2.6) | 1.53 (0.17) | 1.8 (0.21) g/day | NR | -38.0% (7.7) [24h-PER] | • p=0.003 from baseline to FU (MMF) |
| | Conventional therapy (N=20) | 18 | 43.3 (2.8) | 1.65 (0.23) | 1.87 (0.28) g/day | NR | 20.5% (14.1) [24h-PER] | • p=0.351 from baseline to FU (conventional therapy) |
| Lv et al. [74]; NCT00378443 | Prednisone [0.8-1mg/kg/day] + cilazapril [2.5-5mg/day] (N=33) | 48 | 27.8 (8.9) | 1.1 (0.3) | 2.5 (0.9) g/day | 1.04 (0.54) g/day | NR | • p=0.01 between treatment groups |
| | Cilazapril [2.5-5mg/day] (N=30) | 48 | 30.4 (8.8) | 1.1 (0.3) | 2.0 (0.8) g/day | 1.57 (0.86) g/day | NR | |
| Koike et al. [76] | Prednisolone [20-30mg/day] (N=24) | 24 | 37.9 (10.1) | 0.92 (0.26) | 0.68 (0.69) g/day | **0.31 (0.51) g/day** | NR | • p=0.0012 from baseline to FU (prednisolone) |
| | Dipyridamole [150mg/day] or zilazep [300mg/day] (N=24) | 24 | 38.3 (12.7) | 1.15 (0.35) | 0.89 (0.49) g/day | **0.68 (0.69) g/day** | NR | • p=0.2289 from baseline to FU (dipyridamole or dilazep) |
| Lou et al. [77] | Leflunomide [20mg/day] (N=24) | 6 | 29 (11) | NR | 1.66 (0.42) g/day | **0.87 (0.8) g/day** | NR | • P<0.05 from baseline to FU (leflunomide) |
| | Control (N=22) | 6 | 34 (11) | NR | 2.04 (0.46) g/day | 1.63 (0.52) g/day | NR | • P<0.05 from baseline to FU (control) |
| Frisch et al. [78] | MMF (N=17) | 14.75[k] | 39 (19, 72)[k] | 2.6 (1.2) | 2.7 (1.6) g/day | 2.7 (2.3) g/day | NR | • p=0.92 between study arms |
| | Placebo (N=15) | 18[k] | 37 (22, 59)[k] | 2.2 (0.72) | 2.7 (1.4) g/day | 2.5 (NR) g/day | NR | |
| Maes et al. [79] | MMF [1000mg bid] (N=21) | 36 | 39 (11) | NR | 1.9 (0.3) g/day[l] | 1.6 (0.6) g/day[l] | NR | • p=0.0001 between study arms |
| | Placebo (N=13) | 36 | 43 (15) | NR | 1.3 (0.4) g/day[l] | 1.0 (0.6) g/day[l] | NR | |

(Continued)

**Table 7.** (Continued)

| Author | Treatment (N) | FU, mo[a] | Age, yrs[b] | Baseline SCr, mg/dL[b] | Baseline Proteinuria[bc] | FU Protein-uria[bc] | Change from baseline[b] | p-value |
|---|---|---|---|---|---|---|---|---|
| Pozzi et al. [80] & Pozzi et al. [81] | MP [1 g] + prednisone [0.5 mg/kg] + supportive therapy (diuretics, antihypertensive and antiplatelet agents) (N=43) | <120 | 38 (26, 45)[h] | 97.2 (79.6, 114.9) μmol/L[h] | 2.0 (1.6, 2.4) g/day[h] | **0.8 (0.6, 1.3) g/day**[h] | NR | • NR |
| | Supportive therapy (diuretics, antihypertensive and antiplatelet agents) (N=43) | <120 | 40 (29, 51)[h] | 88.4 (79.6, 114.9) μmol/L[h] | 1.8 (1.4, 2.4) g/day[h] | 1.7 (1.1, 3.0) g/day[h] | NR | |
| Katafuchi et al. [82] & Katafuchi et al. [83] | Prednisolone [7.5-20 mg] (N=43) | 60 | 33.6 (13.4) | 0.92 (0.24) | 2.2 (2.0) g/g | 1.34 (1.54) g/g | -0.84 (1.78) g/g | • p=0.002 from baseline to FU (prednisolone) |
| | Control (N=47) | 60 | 32.5 (10.8) | 0.91 (0.21) | 1.1 (0.9) g/g | 0.82 (0.69) g/g | 0.26 (1.65) g/g | • p=NS from baseline to FU (control) |
| | Prednisolone [7.5-20 mg] (N=43) | 24 | 33.6 (13.4) | 0.9 (0.2) | 2.31 (2.01) g/g | 1.70 (1.96) g/g | NR | • p=significant from baseline to FU (prednisolone) |
| | Control (N=45) | 24 | 32.4 (11.1) | 0.9 (0.2) | 1.1 (0.8) g/g | 1.1 (1.30) g/g | NR | • p=NS from baseline to FU (control) |
| Ballardie and Roberts [84] | Prednisolone [40mg/day] + cyclophosphamide [1.5 mg/kg/day] + azathioprine [1.5 mg/kg/day] (N=19) | 48 | >45 | NR | 3.9 (0.8) g/day[i] | **0.8 (0.3) g/day[i]** | NR | • p<0.02 from baseline to FU (treatment). |
| | Control, no immunosuppression (N=19) | 36 | | NR | 4.57 (0.4) g/day[i] | 4.17 (2.8) g/day[i] | NR | • p=NS from baseline to FU (control). |

*(Continued)*

| Author | Treatment (N) | FU, mo[a] | Age, yrs[b] | Baseline SCr, mg/dL[b] | Baseline Proteinuria[bc] | FU Proteinuria[bc] | Change from baseline[b] | p-value |
|---|---|---|---|---|---|---|---|---|
| Locatelli et al. [85] | MP [1 g] + prednisone [0.5 mg/kg] (N=43) | 72 | NR | NR | 2.0 (0.6) g/day | **0.67 (0.5) g/day** | NR | • p=significant from baseline to FU (MP + prednisone) |
| | Supportive treatment (diuretics, antihypertensive drugs and antiplatelet agents) (N=43) | 72 | NR | NR | 1.9 (0.7) g/day | 1.48 (1.87) g/day | NR | |
| Lai et al. [86] | Prednisolone/ prednisone [40-60 mg/day] (N=17) | 37 | 28.9 (7.9) | 115.3 (49.7) µmol/L | 6.5 (2.8) g/day | 2.3 (2.2) g/day | NR | • p<0.001 from baseline to FU (prednisolone/ prednisone) |
| | No corticosteroid therapy (N=17) | 38 | 26.9 (8.6) | 125.5 (54.0) µmol/L | 4.7 (1.4) g/day | 3.3 (2.1) g/day | NR | • p<0.05 from baseline to FU (no corticosteroid therapy) |

**Abbreviations**: CSA, cyclosporine A; eGFR, estimated glomerular filtration rate; FU, follow-up; IQR, interquartile range; MMF, mycophenolate mofetil; mo, months; MP, methylprednisolone; NR, not reported; NS, not significant; PCR, urine protein creatinine ratio; RAS, renin-angiotensin-system; SCr, serum creatine; SD, standard deviation; SEM, standard error of the mean; TRF, targeted-release formulation; yrs, years.

[a]Follow-up durations are presented in months and have been calculated into months and have been calculated into months (4 weeks/ month; 12 months/ year). Follow-up refers to the final and longest duration of time reported or the duration at which the authors presented the change from baseline. The percentage change from baseline is presented for this follow-up duration; [b]Presented as mean (SD) unless otherwise stated; [c]Proteinuria was reported as PCR or 24h-PER and are indicated here with units g/g or g/day, respectively; [d]Presented as median (range); [e]Least square mean ratio to baseline (80% CI); [f]Presented as mean (80% CI); [g]Conference abstract; [h]Presented as median (IQR); [i]Presented as mean (95% CI); [j]Least square mean ratio to baseline (SEM); [k]High was eGFR ≥ 60 ml/min per 1.73 m² and low eGFR was between 30 and 59 ml/min per 1.73m²; [l]Presented as mean (SEM).

**Bold** indicates clinically significant 24h-PER (<1.0g/day) at follow-up.

**Table 8. eGFR outcomes in patients treated with immunosuppressant/ immunomodulatory therapies.**

| Author | Treatment (N) | FU[a], mo | Age[b], yrs | Baseline SCr[b], mg/dL | Baseline eGFR[b], ml/min/1.73m² | FU eGFR[b], ml/min/1.73m² | Change in eGFR[bc] | p-value |
|---|---|---|---|---|---|---|---|---|
| Mathur et al. [97]; ENVISION/ NCT04287985 | Sibeprenlimab [2mg/kg] (N=38) | 12 | 41 (25, 71)[d] | NR | 58.0 (35.0, 154.0)[d] | 61.0 (31.0, 143.0)[d,e] | -2.7 (1.8)[f] Slope: -4.1 (1.7)[f] mL/min/1.73m²/year | • NR |
| | Sibeprenlimab [4mg/kg] (N=41) | 12 | 39 (20, 73)[d] | NR | 64.0 (35.0, 133.0)[d] | | 0.2 (1.7)[f] Slope: 0.1 (1.6)[f] mL/min/1.73m²/year | |
| | Sibeprenlimab [8mg/kg] (N=38) | 12 | 42 (23, 72)[d] | NR | 56.0 (34.0, 109.0)[d] | | -1.5 (1.8)[f] Slope: -0.8 (1.6)[f] mL/min/1.73m²/year | |
| | Placebo (N=38) | 12 | 36 (18, 52)[d] | NR | 68.5 (33.0, 116.0)[d] | 62.0 (32.0, 119.0)[d,e] | -7.4 (1.8)[f] Slope: -5.9 (1.7)[f] mL/min/1.73m²/year | |
| Zhang et al. [135] & Trial record www.clinicaltrials.gov [87]; NCT03373461 | Iptacopan [10mg] (N=20) | 6 | 39.2 (12.4) | NR | 66.0 (28.5) | NR | 0.78 (1.98)[f] | • NR |
| | Iptacopan [50mg] (N=19) | 6 | 36.6 (8.4) | NR | 53.8 (22.7) | NR | -2.35 (2.00)[f] | |
| | Iptacopan [100mg] (N=22) | 6 | 36 (13.2) | NR | 67.0 (31.8) | NR | -2.91 (1.36)[f] | |
| | Iptacopan [200mg] (N=26) | 6 | 42.5 (15.8) | NR | 57.9 (28.9) | NR | -1.18 (1.80)[f] | |
| | Placebo (N=25) | 6 | 39.4 (11) | NR | 65.7 (32.6) | NR | -3.17 (1.87)[f] | |
| Barratt et al. [16]; NefIgArd/ NCT03643965 | TRF-B (N=97) | 9 | 44 (25, 69)[g] | NR | 54.9 (46.4, 68.9)[g] | NR | 7% (3, 13)[h] | • p=0.0014, percentage change between groups at FU |
| | Placebo (N=102) | 9 | 43 (23, 73)[g] | NR | 55.5 (45.5, 67.7)[g] | NR | | |
| Lafayette et al. [17]; NefIgArd/ NCT03643965 | TRF-B [16 mg/day] (N=182) | 24 | 43 (36, 50)[g] | NR | 56.14 (45.5, 70.97)[g] | NR | -6.11 (-8.04, -4.11)[i] Time averaged: -2.47 (-3.88, -1.02)[i] Slope: -3.55mL/ min/1.73m²/year | • p<0.0001, difference between time averaged change • p<0.0035, eGFR slope between groups |
| | Placebo (N=182) | 24 | 42 (34, 49)[g] | NR | 55.11 (45.96, 67.74)[g] | NR | -12.00 (-13.76, -10.15)[i] Time averaged: -7.52 (-8.83, -6.18)[i] Slope: -5.37mL/ min/1.73m²/year | |

*(Continued)*

**Table 8.** (Continued)

| Author | Treatment (N) | FU[a], mo | Age[b], yrs | Baseline SCr[b], mg/dL | Baseline eGFR[b] ml/min/1.73m² | FU eGFR[b], ml/min/1.73m² | Change in eGFR[bc] | p-value |
|---|---|---|---|---|---|---|---|---|
| Hou at al.[89]; MAIN/ NCT01854814 | MMF (N=85) | 36 | 35 (8.7) | NR | 50.9 (18.2) | NR | Slope: -1.2 (0.56) mL/min/1.73m² per year | • p<0.001 difference in slope between arms |
| | Supportive care (N=85) | 36 | 38.2 (9.8) | NR | 49.3 (17.7) | NR | Slope: -3.8 (0.57) mL/min/1.73m² per year | |
| Lv et al. [54]; NCT04291781 | Placebo (N=14) | 6 | 38.3 (6.9) | NR | 85.01 (3.73)[f] | 77.68 (7.80)[f] | -5.70 (NR) | • p=0.002, between change from baseline telitacicept 160mg vs placebo |
| | Telitacicept [160 mg] (N=16) | 6 | 35.9 (9.9) | NR | 77.67 (6.36)[f] | 82.00 (6.75)[f] | 4.32 (NR) | • p=0.015, between change from baseline telitacicept 240 mg vs placebo |
| | Telitacicept [240 mg] (N=14) | 6 | 36.8 (8.8) | NR | 75.705 (6.377)[f] | 78.04 (6.00)[f] | 2.34 (NR) | |
| Sun et al. [92] | Supportive care (N=71) | 9 | 39.3 (10.1) | 120.2 (30.6) µmol/L | 86.42 (33.93) | NR | 0.0% (-12.6, 19.3)[g] | • p=0.9, between groups |
| | Supportive care + fluticasone (N=71) | 9 | 39.6 (10.5) | 127.5 (35.7) µmol/L | 84.2 (32.2) | NR | 4.5% (-12.3, 23.1)[g] | |
| Tam et al. [94] & Trial record for NCT02112838 [95] | Placebo (N=25) | 6 | 43.1 (14.8) | NR | 51 (25,104)[d] | 51 (21, 115)[d] | 1.4 (2.0) | • p=NS changes in eGFR |
| | Fostamatinib [100 mg] (N=26) | 6 | 42.3 (14.1) | NR | 50 (20, 109)[d] | 51 (25,120)[d] | 2.0 (1.8) | |
| | Fostamatinib [150 mg] (N=25) | 6 | 40.6 (11.6) | NR | 35 (18, 103)[d] | 37 (13, 93)[d] | -0.9 (1.9) | |
| Han et al. [52]; NCT02981212 | MMF+ corticosteroid (N=24) | 12 | 44.0 (10.6) | NR | 36.5 (NR) | 37.12 (NR) | 0.61 (NR) | • p=0.0058, between groups at FU |
| | Supportive care (N=20) | 12 | 46.1 (7.8) | NR | 32.81 (NR) | 28.36 (NR) | -4.45 (NR) | • p=0.0031, change between arms |
| Liang et al. [91]; NCT02160132 | 1–2-3 MP group (N=34) | 6 | 29.7 (10.5) | 98.4 (38.2) µmol/L | 89.86 (39.47) | 100.41 (37.28) | NR | • p=0.214, between groups |
| | 1–3-5 MP group (N=34) | 6 | 33.8 (9.9) | 105.5 (47.2) µmol/L | 79.92 (31.23) | 87.9 (44.56) | NR | • p=0.01, 1-2-3 MP FU vs baseline |
| | All (N=68) | 6 | 31.74 (10.3) | 102.0 (42.7) µmol/L | 84.89 (35.67) | 94.14 (41.26) | NR | • p=0.045 1-3-5 MP FU vs baseline |
| | | | | | | | | • p=0.001, all patients FU vs baseline |
| Zhang et al [39][j] | Leflunomide + low-dose prednisone (N=59) | 24 | NR | NR | NR | NR | NR | • p>0.05 between groups at follow-up |
| | Prednisone alone (N=49) | 24 | NR | NR | NR | NR | NR | |

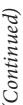

| Author | Treatment (N) | FU[a], mo | Age[b], yrs | Baseline SCr[b], mg/dL | Baseline eGFR[b], ml/min/1.73m² | FU eGFR[b], ml/min/1.73m² | Change in eGFR[bc] | p-value |
|---|---|---|---|---|---|---|---|---|
| Ni et al. [53]; ISRCTN97636235 | Leflunomide [20-40 mg/day] + prednisone [0.5-0.8 mg/kg/day] (N=59) | 24 | 35.7 (11.2) | 99.3 (56.8) μmol/L | 83.9 (39.6) | 83.2 (20.9) | NR | • p=NS from baseline to FU (leflunomide + prednisone)  • p=NS from baseline to FU (prednisone) |
| | Prednisone [0.5-0.8 mg/kg/day] (N=49) | 24 | 35.5 (11.2) | 96.4 (38.6) μmol/L | 84.6 (38.5) | 83.9 (21.5) | NR | |
| Lennartz et al. [55] & Rauen et al. [12, 56]; STOP-IgAN/ NCT00554502 | Single RAS blockade (N=43) | 36 | 45.5 (12.2) | NR | 58.2 (25.7) | 51.1 (29.5) | NR | • NR |
| | Single RAS blockade + immunosuppression (N=39) | 36 | | NR | 62.7 (27.1) | 59.9 (28.0) | NR | |
| | Dual RAS blockade (N=16) | 36 | 44.5 (12.8) | NR | 59.7 (29.8) | 55.3 (38.0) | NR | |
| | Dual RAS blockade + immunosuppression (N=14) | 36 | | NR | 57.4 (18.3) | 53.4 (18.6) | NR | |
| | High eGFR[k] – supportive therapy (RAS blockade) (N=48) | 36 | 45.6 (11.9) | 1.4 (0.5) | NR | NR | -3.78 (13.41) | • p=0.98 between groups at FU |
| | High eGFR[k] – supportive therapy (RAS blockade) + immunosuppression (N=52) | 36 | 41.7 (13.3) | 1.3 (0.4) | NR | NR | -4.07 (15.66) | |
| | Low eGFR[k] – supportive therapy (RAS blockade) (N=23) | 36 | 46.0 (14.0) | 2.0 (0.6) | NR | NR | -5.49 (8.63) | • p=0.90 between groups at FU |
| | Low eGFR[k] – supportive therapy (RAS blockade) + immunosuppression (N=20) | 36 | 45.1 (12.8) | 2.2 (0.7) | NR | NR | -4.64 (9.02) | |
| | Supportive therapy (RAS blockade) (N=71) | 36 | 45.8 (12.5) | 1.6 (0.6) | NR | NR | -4.7 (12.3) | • p=0.32 between groups at FU |
| | Supportive therapy (RAS blockade) + immunosuppression (N=72) [56] | 36 | 42.8 (13.1) | 1.6 (0.7) | NR | NR | -4.2 (14.1) | |
| Liu et al. [57]; NCT02942381 | Hydroxychloroquine sulfate [400 mg/day] (N=30) | 6 | 37.6 (11.6) | 127.9 (41.9) μmol/L | NR | NR | 4.5% (-12.3, 23.1)[g] | • p=0.9 between groups at FU |
| | Placebo (N=30) | 6 | 35.6 (9.6) | 120.2 (32.8) μmol/L | NR | NR | 0% (-12.6, 19.3)[g] | |
| Fellström et al. [58]; NEFIGAN/ NCT01738035 | Placebo (N=50) | 12 | 38.9 (12.0) | NR | NR | NR | -10.9% | • p=0.0134 placebo vs TRF-B [16 mg/day] |
| | TRF-B [8 mg/day] (N=51) | 12 | 40.6 (13.0) | NR | NR | NR | NR | |
| | TRF-B [16 mg/day] (N=48) | 12 | 37.5 (11.9) | NR | NR | NR | 0.7% | |

*(Continued)*

| Author | Treatment (N) | FU[a], mo | Age[b], yrs | Baseline SCr[b], mg/dL | Baseline eGFR[b], ml/min/1.73m² | FU eGFR[b], ml/min/1.73m² | Change in eGFR[bc] | p-value |
|---|---|---|---|---|---|---|---|---|
| Lv et al. [62]; TESTING/ NCT01560052 | MP (N=136) | 36 | 38.6 (11.5) | 1.5 (0.6) | NR | NR | Slope: -1.79 (-4.74, 1.16)[i] | • p=0.03 MP vs placebo change in eGFR |
| Kim et al. [38][j] & Kim et al. [137]; TESTING/ NCT01560052 | Placebo (N=126) | 36 | 38.6 (10.7) | 1.6 (0.6) | NR | NR | Slope: -6.95 (-10.68, -3.21)[i] | |
| Lv et al. [63]; TESTING/ NCT01560052 | MP (N=121) | 12 | 37 (NR) | NR | 65 (NR) | NR | -7.93 (NR) Slope: -0.74mL/ min/1.73m²/year | • p=0.0004, change from baseline in MP group. |
| | Placebo (N=120) | 12 | | NR | | NR | Slope: -3.03mL/ min/1.73m²/year | |
| | MP (N=257) | 42 (median) | 35.6 (29.4, 46.3)[g] | NR | 56.1 (43.2, 75)[g] | NR | Slope: -2.50 (-3.56, -1.44)[i] mL/ min/1.73m²/year | • p=0.002, difference in slope between groups |
| | Placebo (N=246) | 42 (median) | 36.6 (29, 45.9)[g] | NR | 59 (42, 77.6)[g] | NR | Slope: -4.97 (-6.07, -3.87)[i] mL/ min/1.73m²/year | |
| Min et al. [64] | Prednisone [0.8mg/kg/day] (N=45) | 12 | 36.6 (11.5) | 95.1 (31.6) µmol/L | 83.64 (29.51) | 84.14 (26.03) | NR | • p=0.575 between groups at FU |
| | Leflunomide [20-40mg/day] + prednisone [0.8mg/kg/ day] (N=40) | 12 | 36.9 (10.5) | 92.4 (34.0) µmol/L | 83.74 (31.54) | 87.51 (27.66) | NR | |
| Yu et al. [65] & Kim et al. [66]; NCT01224028 | Tacrolimus [0.1mg/kg/day] (N=20) [FU phase][65] | 60 | 36.9 (11.4) | NR | NR | NR | Slope: -6.4 (5.9) ml/min/year | • p=0.637 baseline between groups • p=0.472 FU between groups |
| | Placebo (N=20) [FU phase] | 60 | 41.0 (2.6) | NR | NR | NR | Slope: -5.4 (7.9) ml/min/year | • p=0.143 control vs tac during FU |
| | Placebo (N=20) [treatment phase] | 4 | 40.1 (12.8) | 0.98 (0.26) | 79.6 (21.6) | 77.4 (22.9) | NR | • p=0.988 between groups at FU |
| | Tacrolimus [0.1mg/kg/day] (N=20) [treatment phase] | 4 | 36.9 (11.4) | 1.06 (0.3) | 84.6 (23.2) | 83.1 (24.1) | NR | |
| Hogg et al. [68]; NCT00318474 | MMF [25-36mg/kg/day] (N=7 at FU) | 24 | 31.8 (11.7) | NR | 101.3 (80.6, 109.9)[i] | 84.5 (59.6, 109.4)[i] | -12.6 (-26.6, 1.4)[i] | • p=0.3 mean difference in changes between groups |
| | Placebo (N=10 at FU) | 24 | 32.2 (13.2) | NR | 117.5 (78.8, 156.3)[i] | 90.7 (58.7, 122.7)[i] | -22.0 (-42.0, -2.0)[i] | |
| Liu et al. [69] | MP [0.8mg/kg/day] + CSA [3mg/kg/day] (N=23) | 36.45 (17.08)[b] | 42.4 (13.1) | NR | 80.46 (22.73) | 90.51 (21.41) | NR | • p=0.021 MP + CSA baseline to FU • p=0.004 MP baseline to FU |
| | MP [0.8mg/kg/day] (N=25) | 35.64 (15.74)[b] | 36.8 (8.1) | NR | 81.63 (18.36) | 96.82 (26.55) | NR | • p>0.05 in the changes between the groups |
| Tang et al. [73]; NCT00863252 | MMF [1.5-2g/day] (N=20) | 72 | 42.1 (2.6) | NR | NR | NR | Slope: -1.125ml/ min/1.73 m²/year | • p=0.021 rate of change between groups at FU |
| | Conventional therapy (N=20) | 72 | 43.3 (2.8) | NR | NR | NR | Slope: -3.812ml/ min/1.73 m²/year[i] | |

(Continued)

| Author | Treatment (N) | FU[a], mo | Age[b], yrs | Baseline SCr[b], mg/dL | Baseline eGFR[b], ml/min/1.73m² | FU eGFR[b], ml/min/1.73m² | Change in eGFR[bc] | p-value |
|---|---|---|---|---|---|---|---|---|
| Manno et al. [75] | Ramipril [2.5mg/day] (N=49) | 60[g] | 34.9 (11.2) | 1.07 (0.26) | 97.5 (27.7) | NR | Slope: -6.17 (13.3) ml/min/1.73 m²/ year | • p=0.013 change between groups |
|  | Ramipril [2.5mg/day] + prednisone [1 mg/kg/day] (N=48) | 60[g] | 31.8 (11.3) | 1.08 (0.32) | 100.4 (26.1) | NR | Slope: -0.56 (7.62) ml/min/1.73 m²/ year |  |
| Lou et al. [77] | Leflunomide [20mg/day] (N=24) | 3 | 29 (11) | NR | 77.1 (23.7) | 84.8 (22.6) | NR | • p>0.05 from baseline to FU (leflunomide) |
|  | Control (N=22) | 3 | 34 (11) | NR | 67.0 (28.2) | 66.3 (21.3) | NR | • p>0.05 from baseline to FU (control) |

**Abbreviations**: CI, confidence interval; CSA, cyclosporine A; eGFR, estimated glomerular filtration rate; FU, follow-up; IQR, interquartile range; MMF, mycophenolate mofetil; MP, methylprednisolone; NR, not reported; NS, not significant; RAS, renin-angiotensin system; SD, standard deviation; SEM, standard error of the mean; TRF, targeted-release formulation.

[a]Follow-up durations are presented in months and have been calculated into months (4 weeks/ month; 12 months/ year). Follow-up refers to the final and longest duration of time reported or the duration at which the authors presented the change from baseline. The percentage change from baseline is presented for this follow-up duration; [b]Data is presented as mean (SD) unless otherwise stated; [c]Change from baseline is reported in ml/min/1.73m² unless stated as percentage; [d]Presented as median (range); [e]Data from 3 months follow-up; [f]Presented as mean (SEM); [g]Presented as median (IQR); [h]Presented as percentage change (95% CI) compared to placebo; [i]Presented as mean (95% CI); [j]Denotes conference abstract/ poster; [k]Patients in the high-eGFR arm had eGFR ≥60ml/min/1.73m² at baseline and patients in the low eGFR arm had eGFR between 30 and 59ml/min/1.73 m² at baseline.

in PCR changes compared with the placebo group at follow-up (Table 7). In NefIgArd [16,17], patients treated with TRF-B had a significant reduction in PCR at 9 (-31%) and 24 months (-30.7%) compared with patients in the placebo group (-5%; p=0.0003 and -1%; p<0.0001, respectively; Table 7).

Of the 30 studies reporting 24h-PER, 10 reported a significant reduction from baseline to follow-up in the treatment group (Table 7) and 6 studies reported significantly lower 24h-PER in treatment groups compared to control groups at follow-up (Table 7) [57,62,64,74,120,121], while Mathur et al. [97] did not report statistical significance regarding the alteration in 24h-PER. Treatment with mycophenolate mofetil (MMF) resulted in a significantly higher 24h-PER than placebo plus dietary salt restriction and ACEi at 36-months follow-up [79]. Lafayette et al. [61] reported a non-significant reduction in 24h-PER from baseline to follow-up for patients treated with rituximab plus standard therapy (fish oil with ACEi/ARBs) for 12 months (Table 7). Frisch et al. [78] reported no significant difference in 24h-PER between patients treated with MMF or a placebo at follow-up (Table 7). Patients treated with telitacicept for 6 months in Lv et al. [32] and patients treated with methylprednisolone, prednisone, diuretics, antihypertensives and antiplatelet agents at 10 years follow-up in Pozzi et al. [81] and Pozzi et al. [80] had a substantial decrease in 24h-PER, however significance of the change was not reported in either study (Table 7). Additionally, Tang et al. [72] reported a significant decrease in 24h-PER after treatment with MMF from baseline to 18 months follow-up which was maintained to 72 months follow-up (Table 7) [72].

eGFR was measured in 22 studies (Table 8). Six studies demonstrated that treatment with immunosuppressant/immunomodulatory therapies led to a significantly slower eGFR decline compared to the control/placebo group [14,29,58,62,73,75] and 7 studies showed no significant difference in eGFR decline between treatment and control groups at follow-up (Table 8) [12,56,57,64-66,68,77,94]. From these, eight studies reported data on eGFR slope (Table 8) [17,63,65,66,73,75,89,97]. In the 2023 NefIgArd study, Lafayette et al. found that TRF-B significantly outperformed placebo in treating primary IgA nephropathy (Table 8) [17]. After 2 years, change in eGFR from baseline favored TRF-B, -6.11 mL/min per 1.73 m² in the TRF-B group, −12·00 mL/min per 1·73 m² in the placebo group corresponding to a time-weighted average of eGFR over 2 years of −2·47 mL/min per 1·73 m² for TRF-B and −7·52 mL/min per 1·73 m² for placebo (Table 8; p<0·0001)[17]. Following a similar trend, Hou et al. found that MMF led to a significantly slower eGFR decline (-1.2 (0.56) vs. -3.8 (0.57) mL/min/1.73m²/year, p<0.001) [89]. Lv et al. and Kim et al. provided further evidence, demonstrating that MP reduced eGFR decline (-1.79 vs. -6.95, p=0.03; -2.50 vs. -4.97 mL/min/1.73m²/year, p=0.002) [63]. Yu et al. reported comparable slopes for tacrolimus (-6.4) and placebo (-5.4), and no statistical difference was reported [65,66]. Tang et al. observed a slower decline with MMF (-1.125 vs. -3.812 mL/min/1.73m²/year, p=0.021) [73]. Manno et al. found that the combination of ramipril and prednisone slowed eGFR decline (-0.56) more than ramipril alone (-6.17 mL/min/1.73m²/year, p=0.013) [75]. In the ENVISION study, Mathur et al. reported that the eGFR slope was -4.1, 0.1, and -1.5 mL/min/1.73m²/year in the 2 mg/kg, 4 mg/kg, and 8 mg/kg sibeprenlimab treatment groups, respectively, compared to -7.4 mL/min/1.73m²/year in the placebo group. However, significance was not specified (Table 8) [97]. Overall, each study underscores the efficacy of these treatments in slowing eGFR decline, showcasing varying degrees of effectiveness across different interventions. Notably, treatments such as TRF-B, MMF, and MP at higher doses exhibited significant benefits compared to their respective control or placebo groups.

Furthermore, Tam et al. reported that fostamatinib treatment for 6 months resulted in no significant changes in eGFR from baseline to follow-up (Table 8) [94,95]. Ni et al. [53] reported a non-significant decrease in eGFR for patients treated with prednisone alone or prednisone plus leflunomide (Table 8). One study, Liu et al [69], reported treatment with methylprednisolone with or without cyclosporine A significantly increased eGFR from baseline to follow-up (Table 8).

Mortality and KF rates were low in all studies which reported these outcomes for patients receiving immunosuppression/immunomodulatory therapies (Table 5). Tam et al. [96] reported 1 death in the placebo group (Table 5). AEs were reported in a relatively high proportion of patients receiving immunosuppressive therapies, although discontinuations were not frequently reported. Eight studies reported infections as AEs or serious AEs (SAEs) during immunosuppressive therapy (Table 6) [12,58,60,62,64,95,97,120].

**Table 9. Proteinuria outcomes for patients treated with a combination of therapies.**

| Author | Treatment (N) | FU, mo[a] | Age, yrs[b] | Baseline SCr, mg/dL[b] | Baseline Proteinuria[bc] | FU Proteinuria[bc] | Change from baseline[d] | p-value |
|---|---|---|---|---|---|---|---|---|
| Shima et al. [104]; C000000363 | Prednisolone [2mg/kg/day] + mizoribine [4mg/kg/day] + warfarin [per day] + dipyrida-mole [6mg/kg/day] (N=34) | 24 | NR | 0.5 (0.4, 0.69)[e] | 1.56 (0.66, 2.76) g/g[e] | 0.15 (0.05, 0.47) g/g[e] | NR | • p<0.0001 baseline to FU (prednisolone + mizoribine + warfarin + dipyridamole) |
| | Prednisolone [2mg/kg/day] + mizoribine [4mg/kg/day] (N=36) | 24 | NR | 0.5 (0.4, 0.62)[e] | 1.7 (0.9, 2.82) g/g[e] | 0.21 (0.08, 0.49) g/g[e] | NR | • p<0.0001 baseline to FU (prednisolone + mizoribine) |
| Cheng et al. [106] | Valsartan [80mg/day] (N=42) | 24 | 3.91 (9.72) | 74.62 (19.34) | 2.40 (0.97, 3.41) g/day[e] | 1.50 (0.61, 2.01) g/day[e] | NR | • p=0.045 from baseline to FU (valsartan)<br>• p=0.039 from baseline to FU (valsartan + clopidogrel)<br>• p=0.011 from baseline to FU (valsartan + leflunomide)<br>• p=0.009 from baseline to FU (valsartan + clopidogrel + leflunomide) |
| | Valsartan [80mg/day] + clopidogrel [75mg/day] (N=42) | 24 | 34.05 (9.80) | 73.71 (18.47) | 2.31 (0.92, 3.36) g/day[e] | 1.36 (0.64, 2.30) g/day[e] | NR | |
| | Valsartan [80mg/day] + leflunomide [20mg/day] (N=42) | 24 | 33.74 (8.91) | 74.03 (19.64) | 2.52 (0.85, 3.14) g/day[e] | **0.87 (0.49, 2.03) g/day[e]** | NR | |
| | Valsartan [80mg/day] + clopidogrel [75mg/day] + leflunomide [20mg/day] (N=42) | 24 | 32.92 (8.74) | 74.85 (19.09) | 2.48 (0.88, 3.25) g/day[e] | **0.73 (0.41, 1.68) g/day[e]** | NR | |
| Ye et al. [99]; NCT00426348 | Probucol [750mg/day] + valsartan [160mg/day] (N=34) | 36 | 34 (18, 74)[f] | 105.48 (40.12) µmol/L | 1.39 (0.54) g/day | 1.39 (0.99) g/day | NR | • p=0.99 from baseline to FU (probucol + valsartan)<br>• p=0.66 from baseline to FU (valsartan) |
| | Valsartan [160mg/day] (N=35) | 36 | 34 (19, 67)[f] | 108.81 (50.58) µmol/L | 1.47 (0.76) g/day | 1.34 (0.94) g/day | NR | |
| Kamei et al. [108] & Yoshikawa et al. [110] | Prednisolone [2mg/kg] + azathioprine [2mg/kg/day] + heparin-warfarin + dipyridamole [5mg/kg/day] (N=40) | 24 | 12.2 (3) | NR | 1.35 (1.01) g/day | **0.22 (0.31) g/day** | NR | • p<0.0001 from baseline to FU (Prednisolone + azathioprine + heparin-warfarin + dipyridamole)<br>• p<0.0001 from baseline to FU (Heparin-warfarin + dipyridamole) |
| | Heparin-warfarin + dipyridamole [5mg/kg/day] (N=38) | 24 | 11.6 (2.3) | NR | 1.02 (1.00) g/day | **0.88 (1.34) g/day** | NR | |

(Continued)

| Author | Treatment (N) | FU, mo[a] | Age, yrs[b] | Baseline SCr, mg/dL[b] | Baseline Proteinuria[bc] | FU Proteinuria[bc] | Change from baseline[d] | p-value |
|---|---|---|---|---|---|---|---|---|
| Xie et al. [107]; CRG030600070 | Mizoribine [200-250 mg/day] + losartan [100 mg/day] (N=34) | 12 | 33.68 (10.29) | 84.47 (32.65) µmol/L | 1.21 (0.56) g/day | NR | -0.43 (0.25) g/day | • p<0.01 from baseline to FU (mizoribine + losartan)<br>• p<0.01 from baseline to FU (mizoribine)<br>• p<0.01 from baseline to FU (losartan)<br>• p<0.01 mizoribine + losartan vs losartan<br>• p=NS mizoribine vs losartan |
| | Mizoribine [200-250 mg/day] (N=35) | 12 | 33.63 (11.71) | 79.21 (21.88) µmol/L | 1.35 (0.74) g/day | NR | -0.51 (0.28) g/day | |
| | Losartan [100 mg/day] (N=30) | 12 | 33.67 (11.62) | 77.88 (22.83) µmol/L | 1.12 (0.54) g/day | NR | -0.68 (0.56) g/day | |
| Yoshikawa et al. [109] | Prednisolone [2 mg/kg/day] + azathioprine [2 mg/kg/day] + warfarin + dipyridamole [5 mg/kg/day] (N=40) | 24 | 11.5 (3.2) | NR | 1.29 (1.19) g/day | 0.10 (0.15) g/day | NR | • p<0.0001 from baseline to FU (Prednisolone + azathioprine + warfarin + dipyridamole)<br>• p<0.0001 from baseline to FU (prednisolone) |
| | Prednisolone [2 mg/kg/day] (N=40) | 24 | 11.1 (2.8) | NR | 1.16 (1.13) g/day | 0.12 (0.16) g/day | NR | |
| Chen et al. [102] | Benazepril [10 mg/day] (N=36) | 12 | 31.28 (8.57) | 112.11 (18.51) µmol/L | 1.79 (0.18) g/day | 1.29 (0.17) g/day | NR | • p<0.05 from baseline to FU (benazepril)<br>• p<0.01 from baseline to FU (benazepril + urokinase)<br>• p<0.05 between study arms |
| | Benazepril [10 mg/day] + urokinase (N=35) | 12 | 30.31 (9.4) | 106.95 (21.17) µmol/L | 1.82 (0.27) g/day | 0.62 (0.15) g/day | NR | |
| Woo et al. [113] & Woo et al. [112] | Cyclophosphamide [1.5 mg/kg/day] + dipyridamole [300 mg/day] + warfarin (N=27) | 94 (22) | 25 (6) | 1.2 (0.3) | 2.4 (2.5) g/day | 0.8 (0.8) g/day | NR | • p<0.005 from baseline to FU (treatment)<br>• NS from baseline to FU (control)<br>• p<0.01 from baseline to FU (treatment continuation)<br>• p<0.025 from baseline to FU (control continuation) |
| | Control (N=21) | 86 (22) | 26 (9) | 1.1 (0.2) | 1.7 (2.0) g/day | 2.1 (2.0) g/day | NR | |
| | Continuation of dipyridamole [300 mg/day] + warfarin (N=13) | 104 (9) | 25 (6) | 1.2 (0.3) | 1.7 (1.4) g/day | 0.7 (0.5) g/day | NR | |
| | Control continuation (N=14) | 84 (25) | 26 (9) | 1.1 (0.2) | 3.1 (3.0) g/day | 0.9 (1.0) g/day | NR | |
| | Cyclophosphamide [1.5 mg/kg/day] + dipyridamole [300 mg/day] + warfarin (N=27) [112] | 68 (28) | 25 (7) | 1.1 (0.3) | 2.9 (3.2) g/day | 1.0 (1.2) g/day | NR | • p<0.01 from baseline to FU (treatment)<br>• p=NS from baseline to FU (control) |
| | Control (N=21) | 74 (40) | 24 (5) | 1.3 (0.3) | 1.6 (1.9) g/day | 1.6 (1.6) g/day | NR | |

(Continued)

Table 9. (Continued)

| Author | Treatment (N) | FU, mo[a] | Age, yrs[b] | Baseline SCr, mg/dL[b] | Baseline Proteinuria[bc] | FU Proteinuria[bc] | Change from baseline[d] | p-value |
|---|---|---|---|---|---|---|---|---|
| Walker et al. [114] | Cyclophosphamide [1-2 mg/kg/day] + dipyridamole [100 mg/day] + warfarin (N=25) | 24 | 34.3 (2.4) | 0.1 (0.01) mmol/L | 1.67 (0.35) g/day[g] | 1.15 (0.31) g/day[g] | NR | • p<0.01 from baseline to FU (treatment) • p=NS from baseline to FU (control) |
| | Control (N=27) | 24 | 34.4 (1.9) | 0.12 (0.01) mmol/L | 1.76 (0.34) g/day[g] | 1.89 (0.45) g/day[g] | NR | |

**Abbreviations**: FU, follow-up; IQR, interquartile range; mo, months; NR, not reported; NS, not significant; PCR, urine protein creatinine ratio; SCr, serum creatinine; SD, standard deviation; SEM, standard error of the mean; yrs, years.

[a]Follow-up durations are presented in months and have been calculated into months (4 weeks/ month; 12 months/ year). Follow-up refers to the final and longest duration of time reported or the duration at which the authors presented the change from baseline. The percentage change from baseline is presented for this follow-up duration;

[b]Presented as mean (SD) unless otherwise stated; [c]Proteinuria was reported as PCR or 24h-PER and are indicated here with units g/g or g/day, respectively; [d]Change from baseline to last follow-up as mean (SD) unless otherwise stated, units are given with values; [e]Data presented as median (IQR); [f]Presented as median (range); [g]Presented as mean (SEM).

**Bold** indicates clinically significant 24h-PER (<1.0g/day) at follow-up.

**Table 10. eGFR outcomes for patients treated with a combination of therapies.**

| Author (year) | Treatment (N) | FU[a], mo | Age[b], yrs | Baseline SCr[b], mg/dL | Baseline eGFR[b], ml/min/1.73m² | FU eGFR[b], ml/min/1.73m² | Change in eGFR[bc] | p-value |
|---|---|---|---|---|---|---|---|---|
| Wu et al. [105]; ChiC-TR-TRC-10000776 | No leflunomide (N=189) | 6 | NR | NR | NR | NR | -3.47 (-5.04, -1.90)[d] | • p<0.001 from baseline to FU (difference in change from baseline between no leflunomide and leflunomide group) |
| | Leflunomide (N=176) | 6 | NR | NR | NR | NR | 1.90 (0.29, 3.52)[d] | |
| | No clopidogrel (N=182) | 6 | NR | NR | NR | NR | -0.30 (-1.90, 1.29)[d] | • p=0.368 from baseline to FU (difference in change from baseline between no clopidogrel and clopidogrel group) |
| | Clopidogrel (N=183) | 6 | NR | NR | NR | NR | -1.27 (-2.86, 0.33)[d] | |
| Cheng et al. [106] | Valsartan [80 mg/day] (N=42) | 24 | 33.9 (9.72) | 74.6 (19.3) | 98.76 (11.22) | 79.94 (12.17) | NR | • p=0.583 between study arms at baseline<br>• p<0.001 between study arms at FU |
| | Valsartan [80 mg/day] + clopidogrel [75 mg/day] (N=42) | 24 | 34.1 (9.80) | 73.7 (18.5) | 98.80 (12.03) | 81.22 (13.30) | NR | |
| | Valsartan [80 mg/day] + leflunomide [20 mg/day] (N=42) | 24 | 33.7 (8.91) | 74.0 (19.6) | 98.83 (11.65) | 90.43 (14.28) | NR | |
| | Valsartan [80 mg/day] + clopidogrel [75 mg/day] + leflunomide [20 mg/day] (N=42) | 24 | 32.9 (8.74) | 74.9 (19.1) | 98.78 (11.74) | 92.75 (15.03) | NR | |
| Xie et al. [107]; CRG030600070 | Mizoribine [200-250 mg/day] + losartan [100 mg/day] (N=34) | 12 | 33.7 (10.3) | 84.5 (32.7) µmol/L | 91.50 (29.83) | 90.86 (28.65) | NR | • p=0.3469 at baseline between groups<br>• NS baseline to FU in all groups |
| | Mizoribine [200-250 mg/day] (N=35) | 12 | 33.6 (11.7) | 79.2 (21.9) µmol/L | 95.63 (28.31) | 95.62 (21.28) | NR | |
| | Losartan [100 mg/day] (N=30) | 12 | 33.7 (11.6) | 77.9 (22.8) µmol/L | 97.85 (32.87) | 93.57 (27.86) | NR | |
| Cheng et al. [111] | Captopril [12.5 mg/day] (N=12) | 36 | 37.2 (7.0) | NR | NR | NR | Slope: -0.739 (0.304)[e] | • NR |
| | Captopril [12.5 mg/day] + ticlopidine [500 mg/day] (N=19) | 36 | 38.5 (8.7) | NR | NR | NR | Slope: -0.543 (0.274)[e] | |
| | Nadolol [40 mg/day] (N=16) | 36 | 35.8 (9.7) | NR | NR | NR | Slope: -0.556 (0.157)[e] | |

**Abbreviations:** CI, confidence interval; eGFR, estimated glomerular filtration rate; FU, follow-up; IQR, interquartile range; mo, months; NR, not reported; NS, not significant; SCr, serum creatinine; SD, standard deviation; SEM, standard error of the mean; yrs, years.

[a]Follow-up durations are presented in months and have been calculated into months (4 weeks/ month; 12 months/ year). Follow-up refers to the final and longest duration of time reported or the duration at which the authors presented the change from baseline. The percentage change from baseline is presented for this follow-up duration; [b]Data is presented as mean (SD) unless otherwise stated; [c]Change from baseline is reported in ml/min/1.73m² unless stated as percentage; [d]Linear mixed-effects model presented as mean (95% CI); [e]Mean slope eGFR in ml/ min/ month presented as mean (SEM).

## Combination therapies

Nine studies reported proteinuria outcomes (1 reported PCR and 8 reported 24h-PER; Table 9), 4 reported eGFR outcomes (Table 10).

Shima et al. [104] reported that PCR significantly decreased from baseline to follow-up at 24 months in patients treated with either prednisolone plus mizoribine, warfarin and dipyridamole, or prednisolone plus mizoribine (Table 9).

A significant decrease in 24h-PER from baseline to follow-up was reported in 7 studies (Table 9). One study, Ye et al. [99], reported valsartan with or without probucol for 36 months did not result in a significant decrease in 24h-PER (Table 9). In Xie et al. [107], patients in the losartan alone group had a significantly greater reduction in 24h-PER than patients in the mizoribine plus losartan group (Table 9). Chen et al. [102] reported significantly lower 24h-PER in the benzapril plus urokinase than benzapril alone group at follow-up (Table 9).

Four studies measured eGFR (Table 10). Wu et al. [105] reported a study which assessed addition of leflunomide in combination with telmisartan, with or without clopidogrel, over 6 months. An increase in eGFR was reported for the group receiving treatment combination with leflunomide, while the group receiving the treatment combination without leflunomide had eGFR decline during follow-up, where the difference was statistically significant between groups. [105] Additionally, treatment combinations including clopidogrel resulted in a greater decline in eGFR than combinations without clopidogrel, although this difference was not significant between groups (Table 10) [105]. Cheng et al. [106] reported a significantly higher eGFR at follow-up among patients receiving valsartan plus leflunomide compared with valsartan alone (Table 10). In another study, Cheng et al reported that treatment with either captopril plus ticlopidine or nadolol alone for 36 months resulted in a slower eGFR decline than captopril alone, although the significance of this difference was not assessed [111]. Xie et al. [107] reported a non-significant decrease in eGFR from baseline to follow-up at 12 months for patients treated with mizoribine alone, losartan alone or both in combination (Table 10).

Woo et al. [113] reported that 6 of 27 patients (22%) in the treatment group (cyclophosphamide plus dipyridamole and warfarin) progressed to KF, compared with 7 of 21 patients (33%) in the control group (anti-hypotensive and diuretic therapy) in the initial treatment period with 6 additional patients reaching KF in the continuation period where patients in the treatment group continued to receive dipyridamole and warfarin (Table 5) [112,113]. Three studies reported overall AE rates (Table 6).

## Non-immunosuppressive therapies

Five studies assessing non-immunosuppressive therapies reported proteinuria outcomes (2 reported PCR and 4 reported 24h-PER; Table 11),3 reported eGFR outcomes (Table 12).

Treatment with allopurinol for 6 months resulted in an increase in PCR from baseline to follow-up, although the significance of this change was not assessed (Table 11) [100]. Kanjanabuch et al. [101] reported pioglitazone treatment for 4 months significantly decreased 24h-PER from baseline to follow-up and was significantly lower than the placebo group at follow-up (Table 11). Treatment with sodium cromoglycate (SCG) resulted in a non-significant decrease in 24h-PER from baseline to follow-up at 4 months (Table 11) [103]. No significant changes in eGFR from baseline to follow-up were reported for patients treated with dapagliflozin for 36 months [98], or allopurinol for 6 months (Table 12) [100]. Wheeler et al. [98] reported that patients in the dapagliflozin group exhibited a slower decline in eGFR (-3.5ml/min/1.73m² annually) in contrast to those in the placebo group (-4.7ml/min/1.73m² annually). However, statistical significance was not reported. Additionally, Wheeler et al. [98] also reported that 5 of 137 patients (3.6%) in the dapagliflozin and 16 of 133 patients (12%) in the placebo group progressed to KF and there were no deaths during the study. In the 2023 PROTECT trial by Rovin et al., the efficacy and safety of sparsentan compared to irbesartan were assessed in patients with IgA nephropathy [18]. Over two years, sparsentan showed significant benefits, including a reduction in proteinuria at the primary 36-week endpoint (Table 11) [18]. Moreover, individuals treated with sparsentan exhibited a slower decline in eGFR compared

**Table 11. Proteinuria outcomes for patients treated with non-immunosuppressive therapies.**

| Author | Treatment (N) | FU, mo[a] | Age, yrs[b] | Baseline SCr, mg/dL[b] | Baseline Proteinuria[bc] | FU Proteinuria[bc] | Change from baseline[d] | p-value |
|---|---|---|---|---|---|---|---|---|
| Heerspink et al. [22]; PROTECT/ NCT03762850 | Sparsentan (N=202) | 9 | 46.6 (12.8) | NR | 1.3 (0.8, 1.8) g/g[d] | NR | -49.8% (NR) [PCR][e]; 0.59 (0.51, 0.69) [PCR][f] | • P<0.0001 geometric LS mean ratio sparsentan vs irbesartan |
| | Irbesartan (N=202) | 9 | 45.4 (12.1) | NR | 1.2 (0.9, 1.7) g/g[d] | NR | -15.1% (NR) [PCR][e] | |
| Rovin et al. [18]; PROTECT/ NCT03762850 | Sparsentan (N=202) | 27.5 | 46.6 (12.8) | NR | 1.3 (0.8, 1.8) g/g[d] 1.8 (1.2, 2.9) g/day[e] | NR | -42.8% (-49.8, -35.0) [PCR][e] -46.9% (-53.4, -39.5) [24h-PER][e]; 0.6 (0.5, 0.72) [PCR][f] 0.56 (0.47, 0.68) [24h-PER][f] | • NR |
| | Irbesartan (N=202) | 27.5 | 45.4 (12.1) | NR | 1.2 (0.9, 1.7) g/g[d] 1.8 (1.3, 2.6) g/day[e] | NR | -4.4% (-15.8, 8.7) [PCR][e] -5.9% (-17.9, 7.9) [24h-PER][e] | |
| Shi et al. [100]; NCT00793585 | Allopurinol [100-300 mg/day] (N=21) | 6 | 39.7 (10.0) | 1.3 (0.5) | 0.959 (1.046) g/g | 1.219 (1.063) g/g | 31.8% [PCR] | • NR |
| | Control (N=19) | 6 | 40.1 (10.8) | 1.4 (0.5) | 0.836 (0.599) g/g | 1.17 (0.951) g/g | 48.5% [PCR] | • NR |
| Kanjanabuch et al. [101] | Pioglitazone [30 mg/day] (N=21) | 4 | 42.1 (13.6) | 1.62 (1.3, 1.9)[e] | 2.1 (1.6, 2.6) g/day[g] | 1.2 (0.7, 1.7) g/day[g] | NR | • p<0.05 from baseline (pioglitazone) • NS from baseline (placebo) • p<0.05 between treatment arms at FU |
| | Placebo (N=20) | 4 | 41.4 (11.4) | 1.5 (1.1, 1.8)[e] | 2.0 (0.9, 3.1) g/day[g] | 2.1 (1.3, 2.9) g/day[g] | NR | |
| Sato et al. [103] | SCG [1,200 mg/day] (N=15) | 4 | 38.5 (NR) | NR | 2.75 (1.39) g/day | 1.81 (1.10) g/day | NR | • NR |
| | SCG-responders | 4 | 32.8 (NR) | NR | NR | **0.98 (0.38) g/day** | NR | |
| | SCG-non-responders | 4 | 41.4 (NR) | NR | NR | 2.52 (1.13) g/day | NR | |
| | Control (N=15) | 4 | 35 (NR) | NR | 2.69 (1.49) g/day | 2.84 (1.79) g/day | NR | |

**Abbreviations:** eGFR, glomerular filtration rate; FU, follow-up; IQR, interquartile range; LS, least square; mo, months; NR, not reported; NS, not significant; PCR, urine protein creatinine ratio; SCG, sodium cromoglycate; SCr, serum creatinine; SD, standard deviation; SEM, standard error of the mean; yrs, years.

[a]Follow-up durations are presented in months and have been calculated into months (4 weeks/ month; 12 months/ year). Follow-up refers to the final and longest duration of time reported or the duration at which the authors presented the change from baseline. The percentage change from baseline is presented for this follow-up duration; [b]Presented as mean (SD) unless otherwise stated; [c]Proteinuria was reported as PCR or 24h-PER and are indicated here with units g/g or g/day, respectively; [d]Change from baseline to last follow-up as mean (SD) unless otherwise stated, units are given with values; [e]Data presented as median (IQR); [f]Presented as median (range); [g]Geometric LS mean (95% CI). [d]Presented as median (IQR); [e]Presented as median (95% CI); [f]Presented as LS mean ratio between treatment and placebo at follow-up; [g]Presented as median (range).

**Bold** indicates clinically significant 24h-PER (<1.0 g/day) at follow-up.

**Table 12. eGFR outcomes in patients treated with non-immunosuppressive therapies.**

| Author/ FU[a] | Treatment (N) | FU, mo | Age, yrs | Baseline SCr, mg/dL | Baseline eGFR[b] | Follow-up eGFR[b] | Change in eGFR[c] | p-value |
|---|---|---|---|---|---|---|---|---|
| Rovin et al. [18]; PROTECT/ NCT03762850 | Sparsentan (N=202) | 27.5 | 46.6 (12.8) | NR | 56.8 (24.3) | 51.2 (25.3) | -5.8 (-7.4, -4.2)[d] Chronic slope: -2.7 (-3.4, -2.1)[d] mL/min/1.73m²/ year Total slope: -2.9 (-3.6, -2.2)[d] mL/min/1.73m²/year | • P=0.037, difference in chronic slope between groups • p=0.058, difference in total slope between groups |
|  | Irbesartan (N=202) | 27.5 | 45.4 (12.1) | NR | 57.1 (23.6) | 49.7 (25.6) | -9.5 (-11.2, -7.9)[d] Chronic slope: -2.9 (-3.6, -2.2)[d] mL/min/1.73m²/ year Total slope: -3.9 (-4.6, -3.1)[d] mL/min/1.73m²/year |  |
| Wheeler et al. [98]; DAPA-CKD/ NCT03036150 | Dapagliflozin [10mg/day] (N=137) | 36 | 52.2 (13.1) | NR | NR | NR | Slope: -3.5 (0.5)[e] | • NR |
|  | Placebo (N=133) | 36 | 50.1 (13.1) | NR | NR | NR | Slope: -4.7 (0.5)[e] |  |
| Shi et al. [100]; NCT00793585 | Allopurinol [100-300 mg/ day] (N=21) | 6 | 39.7 (10.0) | 1.3 (0.5) | 69.5 (26.5) | 73.2 (34.8) | NR | • p=0.2 allopurinol vs control at FU • p=0.2 allopurinol baseline vs allopurinol FU |
|  | Control (N=19) | 6 | 40.1 (10.8) | 1.4 (0.5) | 63.6 (27.5) | 68.9 (36.6) | NR | • p=0.9 control baseline vs control FU |

**Abbreviations:** CI, confidence interval; eGFR, estimated glomerular filtration rate; FU, follow-up; IQR, interquartile range; mo, months; NR, not reported; NS, not significant; SCr, serum creatinine; SD, standard deviation; SEM, standard error of the mean; yrs, years.

[a]Follow-up durations are presented in months and have been calculated into months (4 weeks/ month; 12 months/ year). Follow-up refers to the final and longest duration of time reported or the duration at which the authors presented the change from baseline. The percentage change from baseline is presented for this follow-up duration; [b]Data is presented as mean (SD) unless otherwise stated; [c]Change from baseline is reported in ml/min/1.73m2 unless stated as percentage; [d]Presented as mean (95% CI); [e]Least mean squares eGFR slopes (SEM) in ml/ min/ 1.73m2/ year.

to those administered irbesartan (Table 12). After 2 years, change in eGFR from baseline favored sparsentan, -5.8 mL/min per 1.73 m² in the sparsentan group, -9.5 mL/min per 1.73 m² in the irbesartan group corresponding to an eGFR chronic slope of -2.7 mL/min per 1.73 m² for sparsentan and -3.8 for irbesartan (Table 12; p=0.037). eGFR total slope was –2.9 mL/min per 1.73 m² for sparsentan and -3.9 mL/min per 1.73 m² for irbesartan (Table 12; p=0.058) [18]. Furthermore, 2 studies reported mortality and KF rates (Table 5) and 3 reported overall AE rates (Table 6).

### Mortality, kidney failure rate, and safety outcomes

Data for KF and mortality outcomes, along with safety data, are summarized in Table 5 and Table 6, respectively. For additional safety information, please refer to the ClinicalTrials.gov ID (NCT number) provided in Table 6.

### Risk of bias assessment

The 76 RCTs selected for this narrative synthesis generally provided high-quality evidence due to the RCT design and a larger study cohort (≥ 30 total patients). The risk of bias in the RCTs is summarized in Fig 3. Many of the remaining trials which met the PICOS inclusion criteria but were excluded from narrative synthesis, had single-arm design or small population sizes, and therefore have a higher risk of bias than the RCTs selected for narrative synthesis. Risk of bias assessment for all included trials is summarized in S5 and S6 Tables.

## Discussion

This SLR provides a comprehensive overview of the efficacy of pharmacological therapies for IgAN. Previous SLRs have focused on specific treatment classes [122-124] or included a broader range of kidney diseases [125,126], while the current SLR includes any pharmacological treatments assessed in IgAN-specific trials over the past 4 decades.

### Summary of results

A total of 183 studies reported in 254 references were identified for inclusion. After excluding studies with a focus on Chinese traditional medicine, dietary interventions, non-pharmacological treatments, and those that were non-randomized or included fewer than 30 patients, 76 studies (100 references) were selected for narrative synthesis. These additional criteria were applied to select for the studies with comparable interventions and lowest risk of bias.

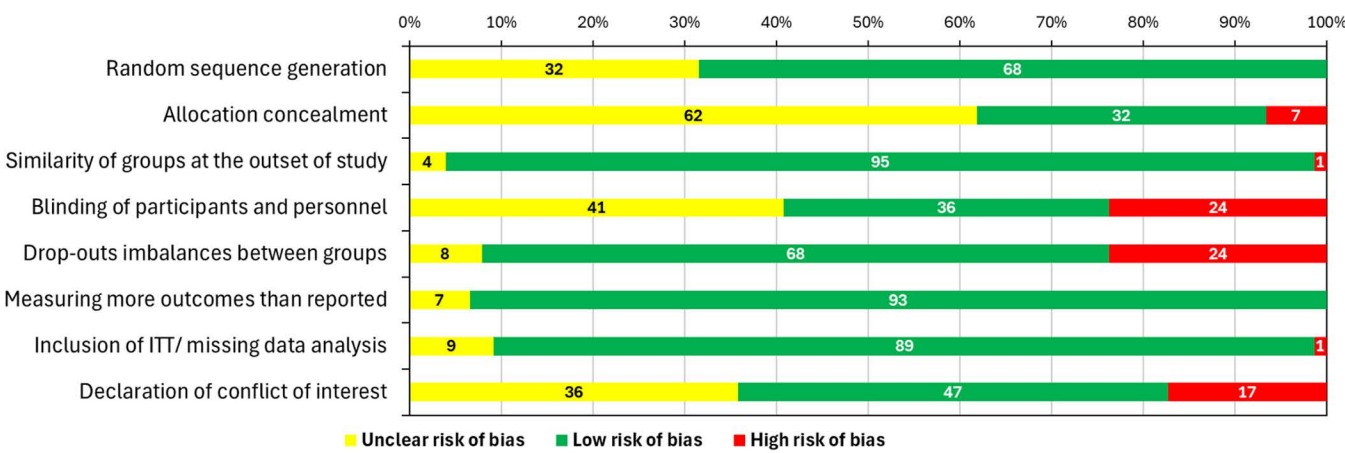

**Fig 3. Risk of bias in the randomized controlled trials.** Abbreviations: ITT, intention to treat.

The findings of this review are consistent with an SLR published in 2003 and updated in 2020 which reviewed literature reporting the effect of immunosuppressive therapies on IgAN [123]. The authors concluded with moderate certainty that corticosteroid therapies may be effective in preventing eGFR decline, although they also noted that more robust evidence from larger trials with a lower risk of bias is required. A more recent SLR, Feng et al. [122] which assessed the clinical outcomes associated with immunosuppressant or corticosteroid therapies was in agreement with Natale et al., [123] concluding more high-quality studies are required to fully understand the efficacy of treatments for IgAN.

## Supportive therapies

The standard of care for IgAN typically includes initial supportive therapy with an ACEi or ARB, either alone or in combination, with the aim of lowering blood pressure leading to a reduction in proteinuria [6]. In the eighteen studies included in this review that investigated treatment of IgAN with supportive therapies, two reported significant reductions in PCR [42,48], ten studies reported significant reductions in 24h-PER and six studies reported maintenance of eGFR over the course of the respective studies [40,42,44,45,59,60,119]. Additionally, a further two studies demonstrated that some supportive therapies are significantly more effective in slowing eGFR decline than other supportive therapies [116] or a placebo [43]. Supportive therapies had a relatively good safety profile with few deaths and adverse events reported in the included studies (Tables 5 and 6). While these supportive therapies can be effective in slowing the decline in kidney function typified in IgAN, they do not offer solutions to the underlying disease.

## Immunosuppressive and immunomodulatory therapies

Patients with IgAN can be treated with Immunosuppressive or immunomodulatory therapies to dampen the immune response which leads to glomerular damage. Along with supportive therapies immunosuppression with steroids is recommended in patients with high risk of progression to kidney failure [6]. In addition to covering long-standing immunosuppressive treatments, this review also included data on recently developed immunomodulatory therapies [33,96,97], including TRF-B [16]. Thirty-eight studies in this SLR reported treatment with immunosuppressive or immunomodulatory therapies. Five studies reported a significant reduction in PCR [14,67,82,83,97], ten reported a significant reduction in 24h-PER (Table 7) from baseline to follow-up and six reported a significantly slower eGFR decline than the respective comparator groups [14,29,58,62,73,75] when treated with immunosuppression or immunomodulatory therapies.

A key measure of decline in kidney function, eGFR slope, was reported in eight studies investigating immunosuppressive or immunomodulatory therapies to treat IgAN (Table 8) [17,63,65,66,73,75,89,97]. Lafayette et al. [17] (NefIgArd), Lv et al. [62] and Kim et al. [38] (TESTING), Hou et al. [89] (MAIN) and Tang et al. [73] reported data for TRF-B, high-dose MP, MMF and MMF, respectively, and showed significantly slowed eGFR decline compared to the placebo or control used in each study (Table 8). Manno et al. [75] highlighted that the combination of ramipril and prednisone resulted in a slower eGFR decline compared to ramipril alone and Mathur et al. [97] (ENVISION) reported that sibeprenlimab at 4 mg/kg nearly stabilized the eGFR slope versus placebo although neither reported significance. Sibeprenlimab has been granted breakthrough designation for treatment of IgAN in the US, subsequent to the favorable outcomes observed in the ENVISION trial [97]. Overall, these studies collectively emphasize the efficacy of various treatments in mitigating eGFR decline, highlighting the significant benefits of interventions such as TRF-B, MMF, and high-dose MP when compared to control or placebo groups, along with promising results of sibeprenlimab in the ENVISION study. However, most patients likely receive supportive therapy or immunosuppressants with systemic effects, in line with current clinical guidelines [6]. Liu et al., [69] did not provide data on the eGFR slope; however, this study uniquely documented a significant increase in eGFR from baseline to follow-up, which occurred during treatment with MP or MP plus cyclosporine A over approximately 3 years.

Studies summarized in this review show that immunosuppressive/corticosteroid therapies are associated with AEs in a relatively high proportion of patients, although these did not frequently lead to discontinuation of the study drug during

the shorter-term clinical trials (Table 6). However, infections were reported in several trials among patients receiving these therapies [12,58,60,62,64]. Systemic immunosuppressive therapies are known to be associated with an increased risk for infections, fractures, and other adverse effects, and KDIGO guidelines recommend caution in some groups of patients for whom the risks may outweigh clinical benefits [6].

### Combination therapies

Eleven studies reported investigation of a combination of therapies, typically including at least one supportive therapy or immunosuppressive therapy. In one study, PCR were significantly reduced over the study period and in seven studies 24h-PER were significantly reduced over the study period (Table 9) in groups receiving combination therapies. One study, Xie et al. [107], reported that eGFR was maintained for 12 months following treatment with mizoribine alone, losartan alone or both in combination (Table 10). Rates of AEs were relatively low, where reported in studies of combination therapies (Table 6). Woo et al. [113], however, reported a high rate of progression to KF in both treatment and control group, likely due to the long follow-up period (up to 104 months) capturing the long term decline in kidney function.

### Non-immunosuppressive therapies

In total five studies investigated non-immunosuppressive therapies (Table 11 and 12). The most impactful of these studies, PROTECT, reported treatment of IgAN with sparsentan, a novel dual endothelin-1 angiotensin II receptor antagonist [18-22]. Noteworthy findings from the 2-year study period indicated significant advantages of sparsentan. Specifically, sparsentan demonstrated a significant reduction in proteinuria compared to irbesartan at the primary 36-week endpoint (Table 11) [18] and a significantly slower decline in eGFR than those receiving irbesartan (Table 12). Where reported, the rate of progression to KF, AEs and discontinuations were similar in the study of drug and placebo or control groups in studies investigating non-immunosuppressive therapies (Tables 5 and 6). Further studies are currently ongoing investigating non-immunosuppressive therapies that have not reported outcomes at the time of this review (Table 13). Several clinical trials exploring non-immunosuppressive options included B-cell modulation to reduce IgA-immune complex generation [127-129], targeting of gut associated lymphoid tissue to reduce production of poorly o-glycosylated IgA1 [14,58], inhibition of IgA1 [130] and regulation of IgAN inflammation [94,95].

### Changes in eGFR in IgAN treatment

Across all treatment types summarized in this review, most studies demonstrated modest changes in eGFR without statistical significance, 9 studies reported significantly better eGFR results in treatment groups than respective control groups [14,17,18,29,43,58,62,73,75,106,116]. However, evidence of a sustained effect is limited with 3 studies reporting results at 12 months or less [14,29,58].

### Preservation of kidney function in IgAN treatment

Overall, few studies (13 of 76 studies) reported the proportion of patients progressing to KF and few reported deaths (12 of 76 studies). Progression to KF typically occurs over decades [131], meaning the short duration of many clinical trials is likely to be insufficient to adequately assess the impact of treatments on progression to KF. Due to this, the Kidney Health Initiative project recommended the use of proteinuria as a surrogate endpoint for prediction of longer-term kidney outcomes to accelerate approval of new therapies to treat IgAN [132]. Indeed, KDIGO 2021 guidelines describe a reduction of 24h-PER to below 1.0 g/day as a treatment target [6]. Overall, 43 studies included in this narrative synthesis reported a decrease in 24h-PER following treatment. Of these, 24 studies reported 24h-PER <1.0 g/day at follow-up in the study treatment group, including treatments from all classes discussed in this review (bold in Tables 3, 7, 9 and 11). Short-term reductions in proteinuria (PCR or 24h-PER) were reported in 33 studies which followed patients for up to 5 years. At

**Table 13. Ongoing trials of IgAN therapies (as of March 2024).**

| Intervention type | NCT Number | Trial Name | Title | Intervention | Phases | Completion Date |
|---|---|---|---|---|---|---|
| Targeted immunosuppression | NCT04541043 | NefIgArd-OLE | Efficacy and Safety in Patients With Primary IgA Nephropathy Who Have Completed Study Nef-301 (NefIgArd-OLE) | Nefecon (TRF-B) | Phase 3 | May 31, 2024 |
| Anti-APRIL | NCT05248659 | N/A | Phase 2/3 Open-Label Trial of Sibeprenlimab in the Treatment of Immunoglobulin A Nephropathy | Sibeprenlimab | Phase 2/3 | December 28, 2028 |
|  | NCT05248646 | Visionary | Visionary Study: Phase 3 Trial of Sibeprenlimab in Immunoglobulin A Nephropathy (IgAN) | Sibeprenlimab | Phase 3 | December 30, 2026 |
|  | NCT05852938 | BION-1301 | A Study of BION-1301 in Adults With IgA Nephropathy | BION-1301 | Phase 3 | May 8, 2028 |
|  | NCT05799287 | N/A | A Study of Telitacicept in Patients With Primary IgA Nephropathy | Telitacicept | Phase 3 | December 2025 |
|  | NCT05596708 | N/A | Study of Telitacicept in Patients With Refractory IgA Nephropathy | Telitacicept | Phase 2/3 | September 30, 2026 |
| Anti-endothelin therapy | NCT04573478 | ALIGN | Atrasentan in Patients With IgA Nephropathy | Atrasentan | Phase 3 | December 1, 2025 |
|  | NCT05834738 | ASSIST | Randomized, Double-blind, Placebo-controlled, Crossover Study of Atrasentan in Subjects With IgA Nephropathy | Atrasentan | Phase 2 | October 1, 2025 |
|  | NCT05856760 | SPARTACUS | A Study to Investigate Safety and Effect of Sparsentan in Combination With SGLT2 Inhibition in Participants With IgAN (SPARTACUS) | Sparsentan | Phase 2 | September 2024 |
|  | NCT05003986 | EPPIK | Study of Sparsentan Treatment in Pediatrics With Proteinuric Glomerular Diseases | Sparsentan | Phase 2 | June 1, 2025 |
| Complement inhibitor | NCT04578834 | APPLAUSE-IgAN | Study of Efficacy and Safety of LNP023 in Primary IgA Nephropathy Patients | LNP023 (iptacopan) | Phase 3 | January 14, 2025 |
|  | NCT04557462 | N/A | A Rollover Extension Program (REP) to Evaluate the Long-term Safety and Tolerability of Open Label Iptacopan/LNP023 in Participants With Primary IgA Nephropathy | LNP023 (iptacopan) | Phase 3 | January 4, 2029 |
|  | NCT05174221 | N/A | A Study of Mezagitamab in Adults With Primary Immunoglobulin A Nephropathy Receiving Stable Background Therapy | Mezagitamab | Phase 1 | March 23, 2026 |
|  | NCT05125068 | AT-1501 | Safety and Efficacy of AT-1501 in Patients With IgA Nephropathy (IgAN) | AT-1501 (Tegoprubart) | Phase 2 | August 01, 2025 |
|  | NCT03608033 | ARTEMIS-IGAN | Study of the Safety and Efficacy of OMS721 in Patients With Immunoglobulin A (IgA) Nephropathy | OMS721 (Narsoplimab) | Phase 3 | April 01, 2023 |
|  | NCT05065970 | IGNAZ | Clinical Trial to Assess Efficacy and Safety of the Human Anti-CD38 Antibody Felzartamab (MOR202) in IgA Nephropathy | Felzartamab | Phase 2 | January 01, 2024 |
|  | NCT05097989 | N/A | Study of ALXN2050 in Proliferative Lupus Nephritis (LN) and Immunoglobulin A Nephropathy (IgAN) | ALXN2050 (Vemircopan) | Phase 2 | August 24, 2026 |
|  | NCT05824390 | N/A | A Randomized, Controlled Clinical Study of Rituximab in Treatment of Primary IgA Nephropathy | Rituximab | Phase 4 | October 1, 2023 |
|  | NCT05162066 | RENEW | BCX9930 for the Treatment of C3G, IgAN, and PMN (RENEW) | BCX9930 | Phase 2 | July 01, 2023 |
| Antisense oligonucleotide | NCT05797610 | IMAGINA-TION | A Study to Evaluate the Efficacy and Safety of RO7434656 in Participants With Primary Immunoglobulin A (IgA) Nephropathy at High Risk of Progression (IMAGINATION) | RO7434656 | Phase 3 | September 30, 2030 |

present there is a lack of evidence showing these changes are maintained in the long term as only 10 studies were found with a follow-up period beyond 5 years [9,49,64,80-83,85,112,113,116,117]. Several of these studies involved proteinuria as an established surrogate endpoint for IgAN clinical trials; however, the length of time that a reduced level of proteinuria

needs to be maintained to mitigate the long-term risk of disease progression has not been defined [133]. The results of ongoing open-label extension studies will provide a clearer understanding of the long-term safety and efficacy of new therapies. In the recently published PROTECT trial, spanning over 110 weeks, the administration of sparsentan compared to the maximally titrated irbesartan showcased notable reductions in proteinuria and the preservation of renal function among patients with IgA nephropathy [18]. Moreover, findings from the NefIgArd trial indicate that a 9-month treatment regimen with TRF-B led to a clinically significant decrease in the eGFR decline and a lasting reduction in proteinuria compared to the placebo [17]. These results imply that TRF-B might exert a disease-modifying influence on patients with IgA nephropathy [17]. Combined, the results of these trials have significant implications for the treatment of IgA nephropathy. Sparsentan and TRF-B represent the first two targeted therapies for IgA nephropathy, with fewer adverse events than previous therapies and potential for long term preservation of renal function. Additionally, as both therapies utilize different mechanisms of action, patients with IgA nephropathy will have distinct treatment options.

## Study quality and heterogeneity in IgAN

In this review studies of varying quality were included due to the long timeframe of literature considered for inclusion. While there are many trials assessing IgAN treatments, many of these are of relatively low methodological quality and thus were excluded from narrative synthesis. Even with a focus on the highest quality studies from the RCTs discussed in this review, evidence is weak for the efficacy of treatments traditionally used to treat IgAN in reducing proteinuria or stabilizing/improving eGFR over the long term.

The included studies had a high degree of heterogeneity in study population, design and outcome reporting. The populations in the included studies comprised both treatment-naïve and treatment-experienced patients, mixed disease severity (measured in baseline proteinuria and baseline eGFR), varied geographic location/ race and ethnicity of patients and variations in study size. Study design heterogeneity included a wide range of treatment and follow-up periods, treatment assignment (randomized, cross-over, non-randomized) and blinding to treatment assignment. Included studies greatly varied the outcome timepoints, statistical analysis and the type of outcomes reported across studies. These sources of heterogeneity were reduced by introduction of selection criteria for RCTs with 30 or more patients and reporting of proteinuria and eGFR outcomes but were not completely eliminated. This underlying heterogeneity may be a reason for the highly variable treatment outcomes described in this review.

More recent studies have more comparable patient populations and study designs, and they exhibit higher study quality. From 2021 onwards primary analyses from large randomized controlled trials have been reported [16-18,20,22,36,88]. The 2023 NefIgArd study demonstrated TRF-B's significant treatment benefit over placebo, suggesting its efficacy in reducing proteinuria and slowing eGFR decline in primary IgA nephropathy [17]. Similarly, while narrowly missing the total eGFR slope endpoint, the PROTECT trials indicated promising benefits of sparsentan in preserving kidney function among IgA nephropathy patients [18]. Based on these results, TRF-B was the first approved treatment for IgAN in the US in 2021, followed by sparsentan in 2023. The findings of both trials highlight the evolving treatment landscape for IgAN, with TRF-B and sparsentan representing significant advancements in immunosuppressive and non-immunosuppressive therapies.

## Study limitations

As IgAN is a rare disease, this review was designed to identify any clinical trial regardless of population size or study design. This led to the inclusion of many small, non-randomized, and single-arm studies with relatively low methodological quality. The comparability of the studies included in this SLR is therefore limited due to the wide range of study methodologies captured using the PICOS criteria. However, this was mitigated somewhat by additional filtering to specifically focus on RCTs with populations of 30 or more patients reporting key kidney function indicators. Reporting timelines for large SLRs can be a limitation, when ongoing trials are identified and discussed. This SLR restricted inclusion to publications

in English, which may have led to a language bias favoring inclusion of publications from English-speaking countries. Despite this language restriction, a high proportion of the studies discussed in this narrative synthesis were conducted in Asia, including China, Japan, South Korea, Singapore, Hong Kong, and Thailand. Differences in IgAN outcomes have been reported between Asian and Caucasian populations, with an increased risk of progression to KF in individuals of Pacific Asian origin [134], which may limit the applicability of trial results to the regions in which they were conducted. This SLR includes studies and publications spanning over 40 years. In this time, reporting standards for clinical trials, clinical guidelines, methods of assessment, and treatments available for IgAN have evolved.

## Conclusions

Many treatments discussed here are non-targeted and have systemic effects which can limit their use [6]. This SLR shows that some of these treatments may reduce proteinuria and/or maintain eGFR during the relatively short follow-up periods within the identified trials, but evidence is often weak because of short duration and/or small sample size. Additionally, many IgAN trials had a single-arm design or small population size, and therefore a high risk of bias meaning they were excluded from discussion in this review. Recently completed studies with longer follow-up periods in larger populations are reporting results that will inform upcoming clinical treatment guidelines. These and other RCTs with longer follow-up will provide stronger evidence of the efficacy and safety of IgAN therapies.

## Supporting information

**S1 Table.  Ovid Embase search string.**
(DOCX)

**S2 Table.  PubMed search string.**
(DOCX)

**S3 Table.  Cochrane library CENTRAL database and database of Systematic Reviews search strings.**
(DOCX)

**S4 Table.  Study design and population characteristics of selected studies.**
(DOCX)

**S5 Table.  Risk of bias assessment for all included RCTs.**
(DOCX)

**S6 Table.  Risk of bias assessment for non-randomized studies.**
(DOCX)

**S1 Data.  Screening results of all studies identified in the literature search.**
(XLSX)

## Acknowledgements

SLR screening, data extraction, and risk of bias assessment were supported by Katherine McAllister and Jessica Adams, former employees of Genesis Research Group (Newcastle upon Tyne, UK).

## Author contributions

**Conceptualization:** Anushya Jeyabalan, Kenar D. Jhaveri, Martin Bunke, Mark E. Bensink.
**Data curation:** David M.W. Cork.

**Formal analysis:** Jonathon A. Briggs, David M.W. Cork.

**Funding acquisition:** Kenar D. Jhaveri, Mark E. Bensink.

**Investigation:** Kenar D. Jhaveri, Jonathon A. Briggs, David M.W. Cork, Mark E. Bensink.

**Methodology:** Anushya Jeyabalan, Kenar D. Jhaveri, Jonathon A. Briggs, David M.W. Cork.

**Project administration:** Martin Bunke, Jonathon A. Briggs, David M.W. Cork, Mark E. Bensink.

**Resources:** Mark E. Bensink.

**Supervision:** Anushya Jeyabalan, Kenar D. Jhaveri, Jonathon A. Briggs, David M.W. Cork.

**Writing – original draft:** Jonathon A. Briggs.

**Writing – review & editing:** Anushya Jeyabalan, Kenar D. Jhaveri, Martin Bunke, David M.W. Cork, Mark E. Bensink.

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
