## [Decision Letter · Decision Letter 0]

8 Jan 2025

We look forward to receiving your revised manuscript.

Kind regards,

Rajendra Bhimma, PhD

Academic Editor

PLOS ONE

Journal Requirements:

[This work was funded by Travere Therapeutics.]

[AJ has served on a scientific advisory board for Calliditas Therapeutics. KDJ is a founder and co-president of the American Society of Onco-Nephrology; reports consultancy agreements with Secretome, George Clinicals, PMV pharmaceuticals and Calliditas. KDJ reports honoraria from the American Society of Nephrology, the ISN, and UpToDate.com; reports serving on the editorial boards of American Journal of Kidney Diseases, CJASN, Clinical Kidney Journal, Journal of Onconephrology, Kidney International, and Nephrology Dialysis Transplantation; reports serving as Editor-in-Chief of ASN Kidney News and section editor for onconephrology for Nephrology Dialysis Transplantation. MB is a consultant for Travere Therapeutics, Inc. JAB is an employee, and DMWC was an employee, of Genesis Research Group which received compensation from Travere Therapeutics, Inc. for conducting this study. MEB is a consultant for Travere Therapeutics, Inc. and reports an additional consultancy agreement with Amgen, Inc.].

We note that you received funding from a commercial source: [Travere Therapeutics, Inc and Amgen, Inc.]

Within this Competing Interests Statement, please confirm that this does not alter your adherence to all PLOS ONE policies on sharing data and materials by including the following statement: ""This does not alter our adherence to PLOS ONE policies on sharing data and materials.” (as detailed online in our guide for authors http://journals.plos.org/plosone/s/competing-interests ). If there are restrictions on sharing of data and/or materials, please state these. Please note that we cannot proceed with consideration of your article until this information has been declared.

5. In the online submission form, you indicated that your data is available only on request from a third party. Please note that your Data Availability Statement is currently missing the name of the third party contact or institution and contact details for the third party, such as an email address or a link to where data requests can be made. Please update your statement with the missing information.

6. We note that there is identifying data in the Supporting Information file <Jeyabalan et al IgAN manuscript Supplemental Material.docx>. Due to the inclusion of these potentially identifying data, we have removed this file from your file inventory. Prior to sharing human research participant data, authors should consult with an ethics committee to ensure data are shared in accordance with participant consent and all applicable local laws.

-Location data

Additional guidance on preparing raw data for publication can be found in our Data Policy (https://journals.plos.org/plosone/s/data-availability#loc-human-research-participant-data-and-other-sensitive-data ) and in the following article: http://www.bmj.com/content/340/bmj.c181.long .

Please remove or anonymize all personal information (Name/Year/ID), ensure that the data shared are in accordance with participant consent, and re-upload a fully anonymized data set. Please note that spreadsheet columns with personal information must be removed and not hidden as all hidden columns will appear in the published file.

7. As required by our policy on Data Availability, please ensure your manuscript or supplementary information includes the following:

Additional Editor Comments:

Please see comments attached by the reviewers.

Reviewers' comments:

Reviewer's Responses to Questions

**Comments to the Author**

1. Is the manuscript technically sound, and do the data support the conclusions?

Reviewer #1: Partly

Reviewer #2: Yes

2. Has the statistical analysis been performed appropriately and rigorously?

Reviewer #1: N/A

Reviewer #2: Yes

3. Have the authors made all data underlying the findings in their manuscript fully available?

Reviewer #1: Yes

Reviewer #2: Yes

4. Is the manuscript presented in an intelligible fashion and written in standard English?

Reviewer #1: No

Reviewer #2: Yes

Reviewer #1: 1. Introduction:

The introduction is well-written, providing adequate background on IgA nephropathy and its clinical significance. However, the paragraph on existing therapies (ACEi, ARBs, corticosteroids) could be expanded to include more discussion on the limitations of these treatments in the context of IgAN, especially in relation to long-term outcomes. This would better set the stage for the need for new therapeutic interventions.

The discussion on the risks of corticosteroid therapy could be more detailed. For example, providing specific data or references on the adverse effects associated with corticosteroids would strengthen the argument for exploring alternative treatments.

2. Methods:

Study Selection: The methods for study selection are clear, and the authors have used appropriate inclusion and exclusion criteria (e.g., randomization, sample size ≥30). However, the protocol for this SLR was not registered. While this is acceptable, it is important to state why this was not done and how the review maintained transparency and rigor in the absence of registration.

Search Strategy: The search strategy is robust, covering key databases and supplementing electronic searches with manual searches of conference abstracts and clinical trial registries. One potential limitation is the restriction to English-language publications. The authors could mention whether non-English studies were considered or excluded, and how this might affect the generalizability of the findings.

Risk of Bias Assessment: The authors mention using NICE guidelines to assess the risk of bias, but the details of this process could be clearer. For instance, a brief summary of the bias assessment results for included studies would be helpful for readers to understand the quality of the evidence synthesized.

3. Results:

Study Characteristics: The results section is well-organized, and the synthesis of the 76 included randomized controlled trials (RCTs) is comprehensive. However, the authors could better highlight the key findings from the Phase 3 trials (NefIgArd and PROTECT) earlier in the results section to draw attention to the most relevant and high-quality evidence.

Data Presentation: The results are presented clearly with appropriate use of tables and figures to summarize study designs, patient characteristics, and key outcomes. However, the text could be improved by making explicit connections between the tables/figures and the narrative. For example, in Table 8 (eGFR outcomes), it would be helpful to briefly summarize the key findings in the text and highlight the clinical significance of the eGFR changes reported.

Study Heterogeneity: While a narrative synthesis is appropriate for the heterogeneity of the included studies, it may be helpful to briefly address the extent of heterogeneity in study designs, patient populations, and interventions. This would provide readers with a clearer sense of the limitations of the review.

4. Discussion:

Interpretation of Findings: The discussion effectively summarizes the key findings and places them in the context of existing literature. The authors highlight the need for further high-quality studies with longer follow-up periods. This is a critical point, and the authors could expand on the potential reasons for the mixed results in the current body of evidence. For example, how might study design, sample size, or bias have contributed to these inconsistent findings?

Implications for Practice: The implications for clinical practice could be more explicitly stated. For instance, what specific recommendations can be made for clinicians based on the findings of this review, especially regarding newer treatments like TRF-B and sparsentan? A brief discussion of the real-world application of these findings would help strengthen the manuscript.

Limitations: The authors acknowledge several limitations of the included studies, such as small sample sizes, risk of bias, and short follow-up periods. However, a more in-depth exploration of the limitations of the review itself (e.g., potential publication bias, exclusion of non-English studies) would enhance the transparency and rigor of the manuscript.

5. Conclusion:

The conclusion summarizes the key points effectively but could be more concise. The call for further randomized controlled trials with longer follow-up periods is appropriate, but it would be helpful to highlight the specific areas in which additional research is most urgently needed (e.g., the long-term efficacy of TRF-B, sparsentan, or combination therapies).

Minor Comments:

Figures and Tables: Ensure that all figures and tables are clearly labeled and referenced in the text. For example, Table 8 on eGFR outcomes could benefit from a more thorough explanation in the main text about how the data were interpreted.

References: Some references are cited more than once in different parts of the manuscript (e.g., references to TRF-B and sparsentan studies). Ensure consistent citation style and check for duplicates in the reference list.

Language: While the manuscript is generally well-written, there are occasional minor grammatical errors and awkward phrasing. Consider a thorough review for clarity and readability.

Reviewer #2: It is with great pleasure that I review the manuscript entitled "Clinical study outcomes in IgA nephropathy: a systematic literature review and narrative synthesis". It is a well-written manuscript that tackles an important topic in IgAN treatment. I have only a few suggestions:

1) I suggest creating subtopics for each treatment modality in the immunosuppressive/immunomodulatory section of the results for better clarity.

2) I suggest adding the statistical test used in each study related to the provided p-values in the tables, when available.

**Do you want your identity to be public for this peer review?** For information about this choice, including consent withdrawal, please see our Privacy Policy

Reviewer #1: No

Reviewer #2: No

---

## [Author Response · Author response to Decision Letter 1]

27 Mar 2025

Rajendra Bhimma, PhD

Manuscript ID: PONE-D-24-42874

Title: Clinical study outcomes in IgA nephropathy: a systematic literature review and narrative synthesis

Dear Dr. Bhimma,

We thank you and the reviewers for their insightful comments on our manuscript. We have carefully addressed each point raised and believe the revisions have improved the clarity and quality of the manuscript. Below, we provide a detailed response to each comment.

Reviewer #1

1. Introduction:

Comment: The introduction is well-written, providing adequate background on IgA nephropathy and its clinical significance. However, the paragraph on existing therapies (ACEi, ARBs, corticosteroids) could be expanded to include more discussion on the limitations of these treatments in the context of IgAN, especially in relation to long-term outcomes. This would better set the stage for the need for new therapeutic interventions. The discussion on the risks of corticosteroid therapy could be more detailed. For example, providing specific data or references on the adverse effects associated with corticosteroids would strengthen the argument for exploring alternative treatments.

Response: We have expanded the discussion on the limitations of current therapies, particularly focusing on long-term outcomes and the adverse effects of corticosteroid therapy. Specific data and references regarding the risks of corticosteroids have been added to strengthen the argument for exploring alternative treatments.

2. Methods:

o Study Selection:

Comment: The methods for study selection are clear, and the authors have used appropriate inclusion and exclusion criteria (e.g., randomization, sample size ≥30). However, the protocol for this SLR was not registered. While this is acceptable, it is important to state why this was not done and how the review maintained transparency and rigor in the absence of registration.

Response: We acknowledge that the protocol was not registered. A statement has been added to the manuscript explaining this and describing the measures taken to ensure transparency and rigor.

o Search Strategy:

Comment: The search strategy is robust, covering key databases and supplementing electronic searches with manual searches of conference abstracts and clinical trial registries. One potential limitation is the restriction to English-language publications. The authors could mention whether non-English studies were considered or excluded, and how this might affect the generalizability of the findings.

Response: Non-English studies were excluded due to language limitations within the review team. This limitation is now explicitly addressed in the limitations section of the manuscript.

o Risk of Bias Assessment:

Comment: The authors mention using NICE guidelines to assess the risk of bias, but the details of this process could be clearer. For instance, a brief summary of the bias assessment results for included studies would be helpful for readers to understand the quality of the evidence synthesized.

Response: A summary of the risk of bias results has been added for clarity.

3. Results:

o Study Characteristics:

Comment: The results section is well-organized, and the synthesis of the 76 included randomized controlled trials (RCTs) is comprehensive. However, the authors could better highlight the key findings from the Phase 3 trials (NefIgArd and PROTECT) earlier in the results section to draw attention to the most relevant and high-quality evidence.

Response: While we presented the results of NefIgArd and PROTECT in the context of similar studies to avoid bias, we appreciate the suggestion. We have ensured these trials are adequately contextualized without giving undue weight to their findings.

o Data Presentation:

Comment: The results are presented clearly with appropriate use of tables and figures to summarize study designs, patient characteristics, and key outcomes. However, the text could be improved by making explicit connections between the tables/figures and the narrative. For example, in Table 8 (eGFR outcomes), it would be helpful to briefly summarize the key findings in the text and highlight the clinical significance of the eGFR changes reported.

Response: Thank you for the comment. As currently structured, we feel we have done as suggested providing explicit connections between the tables/figures and the narrative. For the example provided, Table 8 is referenced 9 times over 3 paragraphs and 429 words. With the manuscript at almost 6000 words, we feel additional text will not help with the flow and coherence for the reader.

o Study Heterogeneity:

Comment: While a narrative synthesis is appropriate for the heterogeneity of the included studies, it may be helpful to briefly address the extent of heterogeneity in study designs, patient populations, and interventions. This would provide readers with a clearer sense of the limitations of the review.

Response: A brief discussion of study heterogeneity, including sources such as variations in study designs and patient populations, has been added to the discussion section.

4. Discussion:

o Interpretation of Findings:

Comment: The discussion effectively summarizes the key findings and places them in the context of existing literature. The authors highlight the need for further high-quality studies with longer follow-up periods. This is a critical point, and the authors could expand on the potential reasons for the mixed results in the current body of evidence. For example, how might study design, sample size, or bias have contributed to these inconsistent findings?

Response: We have added a discussion on how study design, sample size, and bias may have contributed to mixed findings.

o Implications for Practice:

Comment: The implications for clinical practice could be more explicitly stated. For instance, what specific recommendations can be made for clinicians based on the findings of this review, especially regarding newer treatments like TRF-B and sparsentan? A brief discussion of the real-world application of these findings would help strengthen the manuscript.

Response: We have outlined specific implications for clinical practice, emphasizing the potential role of TRF-B and sparsentan as long-term treatment options.

o Limitations:

Comment: The authors acknowledge several limitations of the included studies, such as small sample sizes, risk of bias, and short follow-up periods. However, a more in-depth exploration of the limitations of the review itself (e.g., potential publication bias, exclusion of non-English studies) would enhance the transparency and rigor of the manuscript.

Response: The limitations section now includes a more in-depth discussion of potential biases, including publication bias and the exclusion of non-English studies.

5. Conclusion:

Comment: The conclusion summarizes the key points effectively but could be more concise. The call for further randomized controlled trials with longer follow-up periods is appropriate, but it would be helpful to highlight the specific areas in which additional research is most urgently needed (e.g., the long-term efficacy of TRF-B, sparsentan, or combination therapies).

Response: Areas for additional research are highlighted throughout the discussion section. Including an additional statement in the conclusion would be repetitive and increase the length thereby conflicting with the suggestion for a more concise conclusion.

6. Minor Comments:

o Figures and Tables:

Comment: Ensure all figures and tables are clearly labelled and referenced in the text.

Response: All figures and tables have been reviewed for clarity and proper referencing.

o References:

Comment: Ensure consistent citation style and check for duplicates.

Response: The reference list has been thoroughly reviewed for consistency and duplicates.

o Language:

Comment: Address minor grammatical errors and awkward phrasing.

Response: The manuscript has been reviewed and revised for grammatical accuracy and readability.Reviewer #2

1. Subtopics for Treatment Modalities:

Comment: Create subtopics for each treatment modality in the immunosuppressive/immunomodulatory section of the results.

Response: Thank you for your suggestion. However, we believe that the current layout, which provides a comprehensive narrative of the therapy class as a whole, is more effective. Creating subtopics for each treatment modality would result in numerous small sections, each with limited evidence, potentially disrupting the flow and coherence of the manuscript.

2. Statistical Tests in Tables:

Comment: Include statistical tests related to p-values in the tables when available.

Response: We appreciate your suggestion. Our tables are designed to focus on summarizing the key results of the studies to maintain clarity and conciseness. This approach ensures that the tables remain straightforward and accessible, allowing readers to quickly grasp the main findings without being overwhelmed by extensive statistical details. If needed, readers can refer to the original publications for detailed statistical methods.

Editorial Comments:

1. Amended Role of Funder: Travere Therapeutics provided financial support for this study. Mark Bensink is a consultant for Travere Therapeutics and on behalf of Travere Therapeutics was involved in the study's design, data collection, analysis, and manuscript preparation. This does not alter our adherence to PLOS ONE policies on sharing data and materials.

2. Amended Competing Interests Statement: AJ reports honoraria from Calliditas Therapeutics. KDJ is a founder and co-president of the American Society of Onco-Nephrology; reports consultancy agreements with Secretome, George Clinicals, PMV pharmaceuticals and Calliditas. KDJ reports honoraria from the American Society of Nephrology, the International Society of Nephrology, and UpToDate.com; reports serving on the editorial boards of American Journal of Kidney Diseases, CJASN, Clinical Kidney Journal, Journal of Onconephrology, Kidney International, and Nephrology Dialysis Transplantation; reports serving as Editor-in-Chief of ASN Kidney News and section editor for onco-nephrology for Nephrology Dialysis Transplantation. MB is a consultant for Travere Therapeutics, Inc. JAB is an employee, and DMWC was an employee, of Genesis Research Group which received compensation from Travere Therapeutics, Inc. for conducting this study. MEB is a consultant for Travere Therapeutics, Inc. and reports an additional consultancy agreement with Amgen, Inc. This does not alter our adherence to PLOS ONE policies on sharing data and materials.

3. Additional Supplementary Data: We have included the Supplementary Data S1 Excel file containing the screening results of the articles identified in the literature search.

We hope these revisions meet the expectations of the reviewers and the editorial team. Thank you for the opportunity to revise and resubmit our manuscript. We look forward to your feedback.

Sincerely,

Mark Bensink PhD

HEOR Lead

Travere Therapeutics, Inc.

---

## [Editor Report · Decision Letter 1]

10 Apr 2025

Clinical study outcomes in IgA nephropathy: a systematic literature review and narrative synthesis

PONE-D-24-42874R1

Dear Dr. Anushya Jayabalan

We’re pleased to inform you that your manuscript has been judged scientifically suitable for publication and will be formally accepted for publication once it meets all outstanding technical requirements.

Kind regards,

Rajendra Bhimma, PhD

Academic Editor

PLOS ONE

Additional Editor Comments (optional):

Thank you for the responses. These will be forwarded to the reviewers for further comment.
---

## [Editor Report · Acceptance letter]

PONE-D-24-42874R1

PLOS ONE

Dear Dr. Jeyabalan,

I'm pleased to inform you that your manuscript has been deemed suitable for publication in PLOS ONE. Congratulations! Your manuscript is now being handed over to our production team.

Kind regards,

on behalf of

Professor Rajendra Bhimma

Academic Editor

PLOS ONE